# Identifying regulators of aberrant stem cell and differentiation activity in colorectal cancer using a dual endogenous reporter system

Sandor Spisak[1,2,10], David Chen[1,3,10], Pornlada Likasitwatanakul [1,4,5,6,10], Paul Doan [1,5,10], Zhixin Li[1,5], Pratyusha Bala[1,5], Laura Vizkeleti [7], Viktoria Tisza[2], Pushpamali De Silva [1], Marios Giannakis [1,4,5,8], Brian Wolpin [1,4,8], Jun Qi [9] & Nilay S. Sethi [1,4,5,8] ✉

Aberrant stem cell-like activity and impaired differentiation are central to the development of colorectal cancer (CRC). To identify functional mediators of these key cellular programs, we engineer a dual endogenous reporter system by genome-editing the *SOX9* and *KRT20* loci of human CRC cell lines to express fluorescent reporters, broadcasting aberrant stem cell-like and differentiation activity, respectively. By applying a CRISPR screen targeting 78 epigenetic regulators with 542 sgRNAs to this platform, we identify factors that contribute to stem cell-like activity and differentiation in CRC. Perturbation single cell RNA sequencing (Perturb-seq) of validated hits nominate SMARCB1 of the BAF complex (also known as SWI/SNF) as a negative regulator of differentiation across an array of neoplastic colon models. SMARCB1 is a dependency and required for in vivo growth of human CRC models. These studies highlight the utility of biologically designed endogenous reporter platforms to uncover regulators with therapeutic potential.

Genetic screening is a powerful tool to identify functional components of biological processes in a specific and unbiased manner. CRISPR-Cas9 technology has been the method of choice and initially deployed to reveal cancer dependencies[1–5]. As such, CRISPR-Cas9 screens in different cell types and conditions have typically focused on viability[6] and drug sensitivity as a phenotypic readout[5,7–12]. For example, a genome-wide genetic screen detected GRB7 as a driver of resistance to MEK inhibitor in KRAS-mutant colon cancer[13]. While viability remains

an important readout, it can also lead to identification of essential genes and/or nonspecific hits[14].

Applying genetic screens to functional pathway readouts allow for identification of factors that regulate disease-specific phenotypes[15,16]. Aberrant activation of WNT signaling, which supports stem cell behavior in the intestines, is a hallmark of CRC initiation. Genetic screens in cancer cell lines expressing an exogenous canonical Wnt/β-catenin reporter identified regulators such as CDK8[17,18]. Since

[1]Department of Medical Oncology, Dana-Farber Cancer Institute, Boston, MA, USA. [2]Institute of Molecular Life Sciences, HUN-REN Research Centre for Natural Sciences, Budapest, Hungary. [3]Temerty Faculty of Medicine, University of Toronto, Toronto, ON, Canada. [4]Department of Medicine, Harvard Medical School, Boston, MA, USA. [5]Broad Institute of Massachusetts Institute of Technology (MIT) and Harvard University, Cambridge, MA, USA. [6]Department of Medicine, Faculty of Medicine Siriraj Hospital, Mahidol University, Bangkok, Thailand. [7]Department of Bioinformatics, Faculty of Medicine, Semmelweis University, 1094 Budapest, Hungary. [8]Gastrointestinal Cancer Center, Dana-Farber Cancer Institute, Boston, MA, USA. [9]Department of Cancer Biology, Dana-Farber Cancer Institute, Boston, MA, USA. [10]These authors contributed equally: Sandor Spisak, David Chen, Pornlada Likasitwatanakul, Paul Doan. ✉e-mail: nilay_sethi@dfci.harvard.edu

exogenous reporters may not fully recapitulate the aberrant rewiring in cancer, an endogenous WNT reporter by knocking in a fluorescent probe into the genomic locus of *CTNNB1* was employed for a genetic screen, revealing unique hits when compared an exogenous (i.e., TOPFLASH) WNT reporter system, including epigenetic regulators, cell cycle factors and G protein-coupled receptor pathway components[19]. These functional screens relied on capturing a loss of signal, which has a greater potential for false positives compared to screens using a gain of signal[20], and a single exogenous or endogenous reporter readout, which may be limited in its ability to faithfully capture pathway activity.

Disrupting the balance between stem cell and differentiation programs is a defining property of CRC[21–23]. As such, genomic alterations that hinder intestinal differentiation, either by activating stem cell-like programs or inactivating pro-differentiation pathways, are central to CRC development. Our group recently identified *SOX9* as a key mediator of impaired differentiation by mediating aberrant stem cell-like activity in CRC[24]. Keratin 20 (*KRT20*) is a well-recognized marker of differentiated intestinal cells that is notably absent in normal stem cells and suppressed in models of cancer initiation[24–28].

In this work, to identify functional regulators of aberrant stem cell-like and differentiation activity in CRC, we engineer single and dual endogenous reporter systems by knocking-in fluorescent probes at *SOX9* and *KRT20* genomic loci, respectively[29]. Applying CRISPR screens to our system, we nominate epigenetic factors that regulate stem cell and differentiation programs in CRC.

## Results

### Development of an endogenous reporter by genome-editing *SOX9* locus

*SOX9* functionally blocks differentiation by activating an aberrant stem cell-like transcriptional program in human CRC[24,30]. *SOX9* suppression induces differentiation and impairs CRC growth. These results implicate a dependency of CRC on specific stem cell programs and inspire therapeutic approaches directed at promoting intestinal differentiation. Based on these findings, we reasoned that genetic perturbations disrupting aberrant stem cell-like signaling and inducing intestinal differentiation will serve as targets for therapeutic development in CRC. To this end, we engineered an endogenous reporter system by introducing a fluorescent probe into genomic loci that readout aberrant stem cell-like (i.e., *SOX9*) and intestinal differentiation (i.e., *KRT20*) activity in CRC cells (Supplementary Data 1). To establish the *SOX9* reporter, we knocked-in a cassette containing GFP and neomycin antibiotic resistance in-frame at the end of the *SOX9* coding region of a CRC cell line using a combination of CRISPR/Cas9 technology and template-based homologous recombination[29] (Fig.1a). Engineered CRC cells were propagated in media containing neomycin to select populations with in-frame integration. Accurate genomic integration was confirmed using site-specific PCR with primers against genomic locus and cassette (Supplementary Fig. 1a). Compared to parental LS180 CRC cells, engineered GFP knock-in LS180 (LS180$^{SOX9-GFP}$) displayed GFP expression (Fig. 1b and Supplementary Fig. 1b). To validate that GFP faithfully reflects SOX9 expression, two constitutively expressed *SOX9* shRNAs were independently introduced into LS180$^{SOX9-GFP}$ cells, which led to a greater than 50% reduction of GFP levels (Fig. 1c and Supplementary Fig. 1c, d). These studies established an endogenous reporter CRC cell line that reflects aberrant stem cell-like activity through GFP expression.

To evaluate the performance of our platform in a pooled genetic perturbation format, we performed sgRNA and shRNA screens using custom libraries consisting of 76 sgRNAs or 154 shRNAs targeting positive and negative control genes, including *SOX9* and *GFP* (Supplementary Data 3). After introducing the libraries and sorting cells based on GFP expression at two different time points, genomic DNA was sequenced to determine sgRNA or shRNA representation in the high, low, and negative GFP fractions

(Fig. 1d). To account for variation in sample and sequencing depth, we normalized by 1) total read count, 2) plasmid library pool, and 3) population before comparing the sorted cell fractions (Fig. S1e). As expected, GFP and *SOX9* sgRNAs and shRNAs were consistently depleted in GFP high fractions compared to GFP low and negative fractions (Fig. 1e, f), validating the *SOX9* reporter line in the genetic screening format. Confirming a consensus among genetic screens, we noted that CRISPR perturbations provided stronger and more consistent discriminatory power compared to shRNA-mediated suppression (Fig. 1e). We also observed that screening performance at 7 to 10 days of culture following library introduction was superior to 3 days of cultures (Fig. 1f).

After validating the platform in a pooled screening format, we next performed a discovery CRISPR screen focused on druggable epigenetic regulators and their family members (Supplementary Data 2)[31–33], asking which genes when perturbed reduce *SOX9* expression. To demonstrate the broad utility of the reporter system, we utilized another CRC cell line and different fluorescent probe. 542 sgRNAs targeting 78 genes associated with epigenetic regulation were introduced into HT29$^{SOX9-mKate2}$ cells (Supplementary Data 3). Cells were sorted into four even quartiles based on mKate2 expression and comparisons of normalized sgRNA abundance was made between each quartile (25–50%, 50–75%, 75–100%) and the quartile with the lowest mKate2 expression (0–25%). As expected, we found that sgRNAs targeting *SOX9* and *mKate2* were depleted from all three (25–50%, 50–75% and 75–100%) quartiles compared to the low (0–25%) mKate2 expressing subpopulation (Fig. 1g).

To estimate the degree of selection after a gene perturbation, we used MaGeCK Maximum Likelihood Estimation (MLE) to generate a beta score based on the differences in normalized sgRNA abundance when comparing mKate2 fractions. We identified several candidate genes, including but not limited to *SUZ12*, *SMARCD2*, *DNMT1*, and *KMT2A*, which when perturbed suppressed SOX9 reporter activity (Fig. 1h). We then applied a rank sum scoring method to identify consistently depleted sgRNAs in the mKate2 high fraction across replicates. We considered genes that had at least 2 or more targeting sgRNA within the bottom 15% percentile of rank sum scores as stronger candidates (Fig. 1i). As expected, sgRNAs targeting *SOX9* and *mKate2* were depleted in the mKate2 high fraction, along with sgRNAs targeting members of the histone deacetylase family (*HDAC2*, *HDAC4*), the sirtuin family (*SIRT1*, *SIRT2*, *SIRT4*), and the SWI/SNF complex (*SMARCA2*, *SMARCB1*, *SMARCD2*, *SMARCE1*). Together, these data support the use of an endogenous *SOX9* reporter system for functional genetic screens that may help identify regulators of aberrant stem cell-like behavior in CRC.

### Endogenous differentiation reporter by genome-editing *KRT20* locus

A principle reason that aberrant stem cell-like activity is selected as an early event in CRC pathogenesis is to block differentiation and prevent cell death[22,24,26]. In other words, aberrant stem cell-like signaling antagonizes intestinal differentiation in colorectal cancer initiation, enabling the persistence of neoplastic colon cells rather than their turnover in a rapidly renewing epithelium. We therefore sought to develop an endogenous reporter that faithfully captures differentiation activity. KRT20 is a well-recognized marker of differentiated intestinal cells[26,27] that is suppressed upon neoplastic initiation as shown in an Apc$^{KO}$ mouse model (Fig. 2a). We integrated the GFP cassette into the *KRT20* genomic locus of HT29 and LS180 CRC cell lines (Fig. 2b). To biologically validate the system, we asked whether disrupting SOX9 induced differentiation reporter activity based on our previous results[24,30]. HT29$^{KRT20-GFP}$ reporter line stably expressing a shRNA against SOX9 led to a shift in the number of GFP+ cells (Supplementary Fig. 2a, 0.3% to 62.4%), confirming that induction of

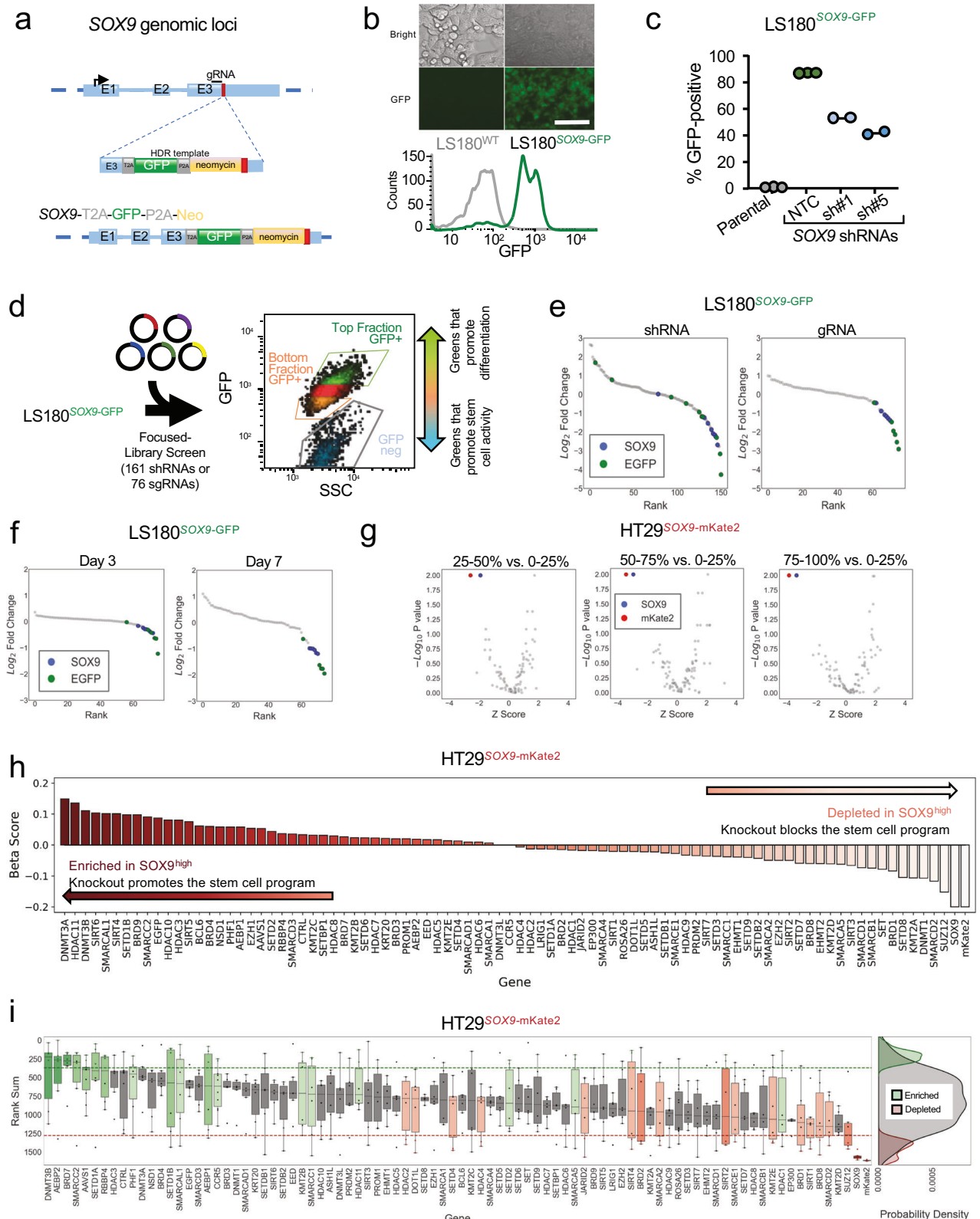

differentiation via *SOX9* suppression is captured by our endogenous differentiation reporter system.

To validate the endogenous KRT20 reporter in a pooled genetic perturbation format, we applied our control shRNA and sgRNA libraries to the engineered cell lines (Supplementary Data 3). After introducing the libraries into HT29^KRT20-GFP cells, we determined shRNA and sgRNA distribution in GFP high versus low sorted fractions using

25% gating. Similar to the endogenous *SOX9* reporter, the endogenous *KRT20* reporter demonstrated greater discrimination with the CRISPR library at day 7 (Fig. 2c, d and Supplementary Fig. 2b, c). As expected, sgRNAs targeting *KRT20* and *GFP* were depleted in GFP high fraction compared to GFP low fraction, providing technical validation, whereas control sgRNAs showed a normal distribution among fractions (Fig. 2c–e). Notably, SOX9 sgRNAs were enriched in the GFP high

**Fig. 1 | Development of an endogenous reporter by genome-editing *SOX9* locus.**
**a** Schematic showing integration of T2A-GFP-P2A-Neo reporter cassette downstream of *SOX9*'s 3rd exon. Stop codon indicated by red box. **b** Light microscopy (top) and histogram (bottom) showing distribution of GFP fluorescence intensity of the LS180$^{SOX9-GFP}$ and its parental cells. **c** Percentage of GFP+ cells in parental and LS180$^{SOX9-GFP}$ upon NTC or SOX9 knockdown. **d** Schematic of 76 sgRNA CRISPR-Cas9 and 161 shRNAs screens targeting 8 genes in the LS180$^{SOX9-GFP}$ line. **e** Ranked log$_2$FC plot of shRNA and CRISPR-Cas9 screens in LS180$^{SOX9}$ line comparing the top and bottom 2.5% GFP positive fractions. SOX9 and EGFP shRNAs or sgRNAs are in blue and green, respectively. **f** Ranked log$_2$FC plot of shRNA and CRISPR-Cas9 screens in LS180$^{SOX9}$ line comparing the top and bottom 2.5% GFP positive fractions on days 3

and 7 following library infection. **g** Volcano plots of CRISPR-Cas9 screen targeting epigenetic regulators (542 sgRNAs targeting 78 genes) in HT29$^{SOX9-mKate2}$ cell-line comparing the indicated 4 mKate2 fractions on day 7. The x-axis shows the $Z$ score of gene-level median log$_2$FC of all sgRNAs while the y-axis shows the gene-level $p$-values generated by MaGeCK MLE. **h** Beta score of each gene in CRISPR-Cas9 screen targeting epigenetic regulators in HT29$^{SOX9-mKate2}$ line (3 technical replicates) comparing the top and bottom 25% mKate2+ fractions. **i** Rank sum of each gene in CRISPR-Cas9 screen targeting epigenetic regulators in HT29$^{SOX9-mKate2}$ line comparing top and bottom 25% of mKate2+ fractions. Boxplots show distribution of sgRNAs per gene. Probability density plots showing the distribution of enriched (green), depleted (red), and all other sgRNA (gray) rank sums (right panel).

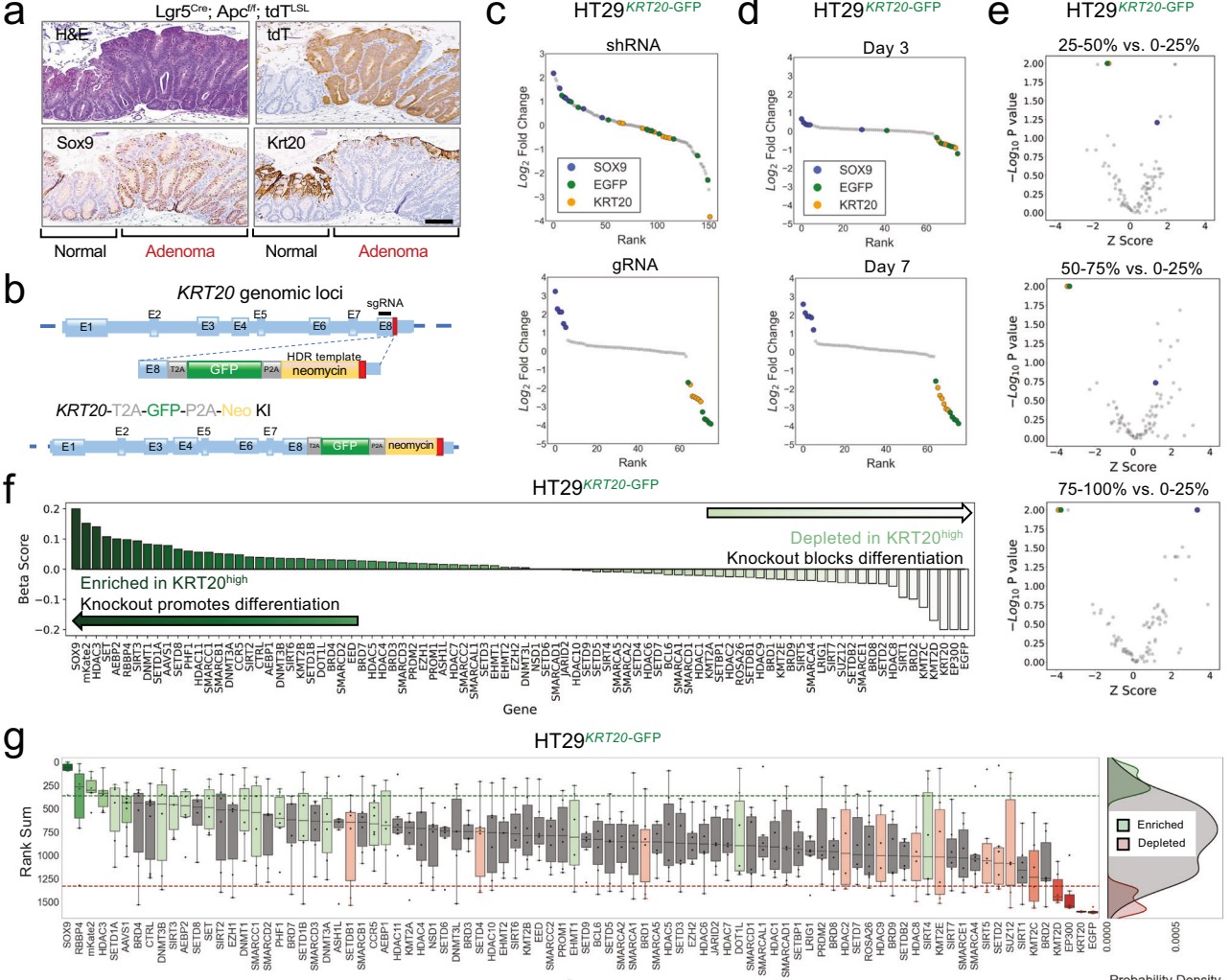

**Fig. 2 | Endogenous differentiation reporter by genome-editing *KRT20* locus.**
**a** H&E as well as Sox9, Krt20, and tdT immunohistochemistry in normal and Apc$^{KO}$ mouse colon; scale bar = 100 μM. **b** Schematic showing integration of T2A-GFP-P2A-Neo reporter cassette downstream of *KRT20*'s 8th exon. Stop codon indicated by red box. **c** Ranked log$_2$FC plot of 161 shRNAs and 76 sgRNAs CRISPR screens in HT29$^{KRT20-GFP}$ cells comparing the top and bottom 2.5% GFP+ fractions. SOX9, KRT20, and EGFP shRNAs or sgRNAs are indicated by blue, yellow, and green, respectively. **d** Ranked log$_2$FC plot of CRISPR-Cas9 screen in HT29$^{KRT20-GFP}$ cells comparing the top and bottom 2.5% GFP+ fractions on days 4 and 7 following library infection. **e** Volcano plots of CRISPR-Cas9 screen targeting epigenetic

regulators (542 sgRNAs targeting 78 genes) in HT29$^{KRT20-GFP}$ cell-line comparing the indicated 4 GFP fractions on day 7. The x-axis shows the $Z$ score of gene-level median log$_2$FC of all sgRNAs while the y-axis shows the gene-level, adjusted $p$-values generated by MaGeCK MLE. **f** Beta score of each gene in CRISPR-Cas9 screen targeting epigenetic regulators in HT29$^{KRT20-GFP}$ line (3 technical replicates) comparing the top and bottom 25% GFP+ fractions. **g** Rank sum of each gene in CRISPR-Cas9 screen targeting epigenetic regulators in HT29$^{KRT20-GFP}$ line comparing top and bottom 25% of GFP+ fractions. Boxplots show distribution of sgRNAs per gene. Probability density plots show distribution of enriched (green), depleted (red), and all other sgRNA (gray) rank sums (right panel).

fraction (Supplementary Fig. 2d, e), providing biological validation that disrupting SOX9 activity induces the differentiation reporter. The enrichment of SOX9 sgRNAs gradually improved by gating a more stringent GFP high population (Supplementary Fig. 2e). These results provide strong evidence that the *KRT20* reporter system is powered to detect genetic perturbations that promote intestinal differentiation of CRC.

We next applied the sgRNA library targeting epigenetic regulators to the HT29$^{KRT20\text{-}GFP}$ reporter cell line. By comparing MLE beta scores of sgRNA abundance in GFP high versus low fractions, we identified several epigenetic regulators that support differentiation upon perturbation, including but not limited to *HDAC3, SET, AEBP2, and RBBP4* (Fig. 2f). We then applied the rank sum scoring method to replicates of the epigenetic regulator screen in HT29$^{KRT20\text{-}GFP}$ cells (Fig. 2g). In addition to confirming our positive and negative controls, we found that perturbations in core histone binding proteins (*RBBP4*), DNA methyltransferase family members (*DNMT3B*), and the histone lysine methyltransferase family members (*SETD1A, SETD1B*) induced differentiation by reporter activity. These results underscore the utility of coupling functional CRISPR screens with endogenous reporters to identify functional regulators of cancer-specific properties.

## Dual stem cell and differentiation reporter system

While the single-reporter systems were successful, we observed considerable variability in individual sgRNA scores (Figs. 1i and 2g) and few genes that once perturbed disrupted both stem cell-like activity and induced differentiation. We therefore sought to improve signal to noise and identify perturbations that induced differentiation by potentially disrupting stem cell-like activity. To this end, we engineered a dual endogenous reporter system to simultaneously monitor aberrant stem cell-like and differentiation programs within the same cell (Fig. 3a). HT29 CRC cells were edited to express mKate2 from the *SOX9* locus and GFP from the *KRT20* locus (HT29$^{SOX9\text{-}mKate2/KRT20\text{-}GFP}$). By applying bulk RNA sequencing to flow-sorted cells, we validated that the SOX9-mK2$^{high}$ fraction demonstrates greater expression of stem cell genes, whereas the KRT20-GFP$^{high}$ fraction demonstrates elevated expression of differentiation genes (Fig. 3b, c). SOX9-mK2$^{high}$/KRT20-GFP$^{low}$ demonstrated elevated expression of stem cell markers *ASCL2, SMOC2* and *LGR5* (Fig. 3c). KRT20-GFP$^{high}$/SOX9-mK2$^{low}$ (relatively differentiated) cells displayed elevated differentiation markers as indicated by increased *KRT20, MUC2* and *DPP4* expression. These findings were further confirmed by gene set enrichment analyses (GSEA) (Fig. 3b). After introduction of our control sgRNA library, we performed two-color flow at day 7, setting the threshold at 2.5% (Supplementary Fig. 3a). We then compared the enrichment of sgRNAs between the mKate2$^{low}$/GFP$^{high}$ fraction and the mKate2$^{high}$/GFP$^{low}$ fractions (i.e., GFP vs. mKate2). We observed that sgRNAs against *SOX9* and *mKate2* were the most enriched, whereas sgRNAs against *KRT20* and *GFP* were the most depleted (Fig. 3d and Supplementary Fig. 3b). Reassuringly, the negative control sgRNAs showed no preference for either bin (Fig. 3d).

We established two additional dual endogenous reporter cell lines, HT115$^{SOX9\text{-}mKate2/KRT20\text{-}GFP}$ and LS180$^{SOX9\text{-}mKate2/KRT20\text{-}GFP}$, confirming utility of this system in other CRC cell lines for functional genetic screens (Supplementary Fig. 3c,d). Notably, the magnitude of sgRNA discrimination was greater in the dual reporter compared to single-reporter system, improving our ability to identify genes that regulate this specific aberrant stem cell-differentiation axis in CRC (Supplementary Fig. 3b compared to Supplementary Fig. 2b). To control for potential technical fluorescent color bias, we engineered dual reporters by swapping fluorescent probes: *GFP* was expressed from the *SOX9* locus and *mKate2* from the *KRT20* locus (Supplementary Fig. 3e-g). Consistent with the original dual reporter CRC cell lines, the ones with reversed fluorescent probes showed the expected distribution of *SOX9, GFP, KRT20*, and *mKate2* sgRNAs relative to controls. These data

demonstrated the successful development of an endogenous dual stem cell-like and differentiation reporter system in CRC.

## Identifying epigenetic regulators of aberrant stem cell-like and differentiation activity in CRC using dual endogenous reporter system

To identify regulators of aberrant stem cell-like and differentiation activity, we applied the sgRNA library targeting epigenetic regulators to the HT29$^{SOX9\text{-}mKate2/KRT20\text{-}GFP}$ reporter cell line given its superior performance in discriminating control sgRNAs (*SOX9, KRT20, mKate2, GFP*) (Supplementary Fig. 3h–j). The top-scoring perturbations that induced differentiation/suppressed aberrant stem-cell activity (potential therapeutic targets) included *SOX9* as well as genes associated with the BAF chromatin remodeling complex (*SMARCA4, SMARCB1, SMARCD1, SMARCA5*), histone lysine methyltransferase (*SET, KMT2A, SETD2, DOT1L, EHMT2, EZH1*), and DNA methyltransferase (DNMT1) (Fig. 3e). Conversely, perturbations that may promote aberrant stem cell-like activity and impair differentiation (cancer-promoting genes) included bromodomain-containing proteins (*BRD1, BRD2*), histone acetyltransferases (*EP300*), and histone lysine methyltransferases (*KMT2C, NSD1, SETD1B*).

We analyzed perturbations that promote differentiation captured by dual and single-channel comparisons, demonstrating that dual-channel comparison (GFP vs. mKate2) consistently identified both control gRNA (*SOX9, mKate2*) as well as exclusive candidate perturbations that induce differentiation compared to either single-reporter system (Fig. 3g, h). Given the biological variability of gRNA knockout efficiency and the superior performance of the dual endogenous reporter system to discriminate candidate hits, we observed that different systems may identify different gRNA perturbation hits that target the same gene. Interestingly, the hits uniquely attributed to the mKate2$^{low}$/GFP$^{high}$ and mKate2$^{high}$/GFP$^{low}$ fraction comparison of the dual reporter system primarily consisted of genes associated with the SWI/SNF chromatin remodeling complex. Indeed, the rank sum scoring method to identify consensus perturbations across the three replicates revealed genes associated with the BAF chromatin remodeling complex (*SMARCA4, SMARCA5, SMARCB1, SMARCC1, SMARCD1*), DNA methyltransferase (*DNMT1*), histone lysine methyltransferase (*DOT1L, SETD2*), lysine methyltransferase (*KMT2A*), and sirtuin (*SIRT4*) (Fig. 3i).

## SMARCB1 restricts intestinal differentiation in CRC

Based on our screen results, we focused on 12 genes that upon perturbation suppressed aberrant stem cell-like and activated differentiation reporter activity. Of note, most of the candidate genes encode proteins that are part of the BAF complex and histone methyltransferase family, save for 2 DNMTs and SIRT4 (Fig. 4a). Using the best performing sgRNAs, we examined whether individual gene knockout (KO) induced KRT20 and/or suppressed SOX9 in HT29 cells by immunoblot (Supplementary Fig. 4a, b). Perturbation of three genes – *KMT2A, SMARCA4*, and *SMARCB1* –increased KRT20 protein expression by immunoblot (Supplementary Fig. 4a, b).

To contextualize the effect of these genetic perturbations on differentiation beyond KRT20 expression, we performed single cell transcriptome analysis coupled with genetic perturbations (Perturb-seq)[34] for our validated hits in HT29 CRC cells. Compared to non-targeting controls, cells with sgRNAs targeting *SOX9* and *CTNNB1* (encodes β-catenin) clustered near one another (Fig. 4b, left) and demonstrated similar transcriptional states involving activation of intestinal differentiation (Fig. 4b, right). While *KMT2A* KO did not meet technical standards, *SMARCA4* and *SMARCB1* perturbation led to distinct cell states with differentiation induction, albeit to a lesser extent than *SOX9* and *CTNNB1* inactivation (Fig. 4b, c). Further analysis revealed that broad multi-lineage differentiation was activated upon

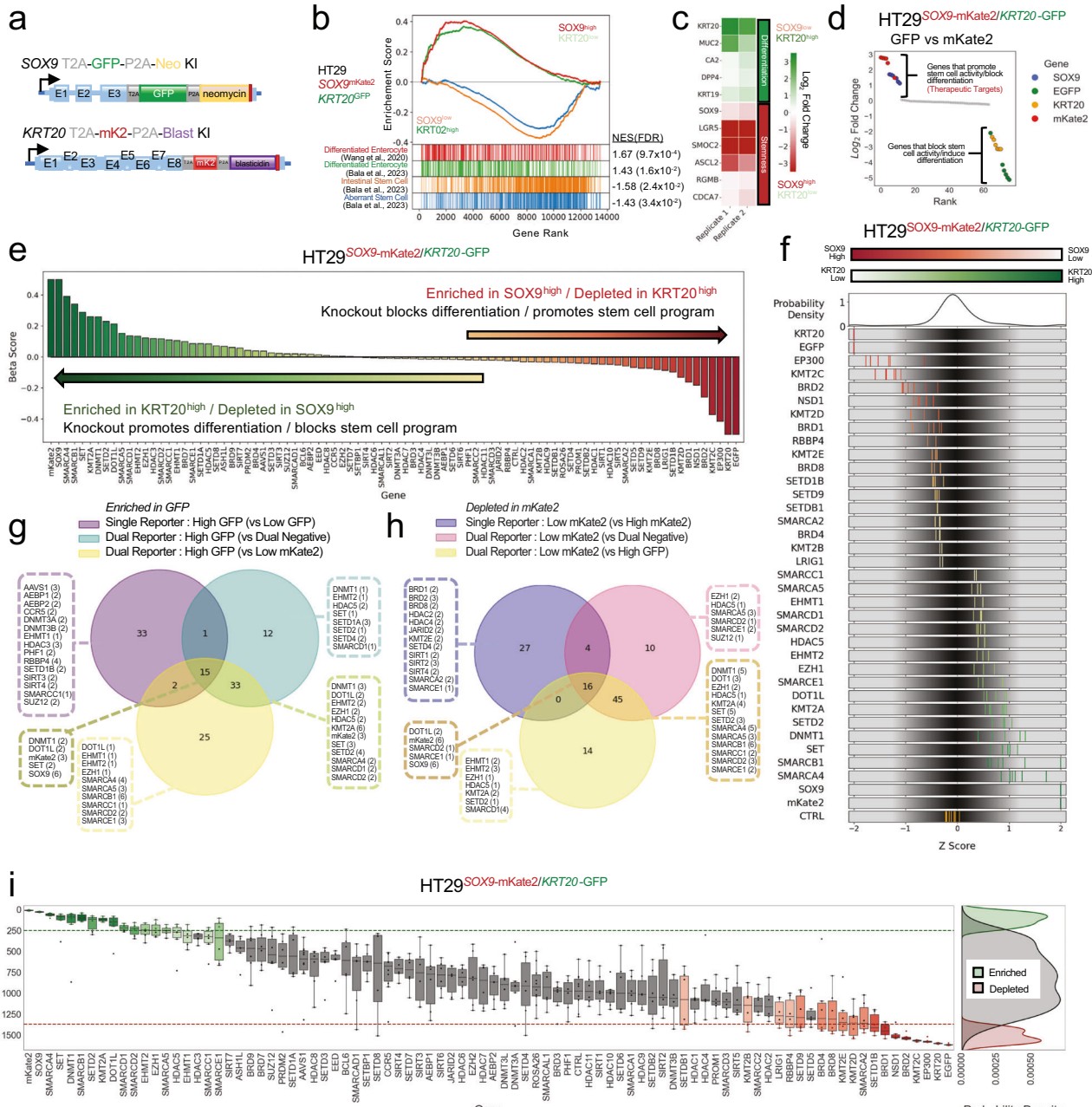

**Fig. 3 | Epigenetic CRISPR screen using dual endogenous reporter system.**
**a** Schematic of engineered *SOX9* and *KRT20* loci in dual endogenous reporter lines following GFP and mKate2 (mK2) HDR template integration, respectively. **b** GSEA of bulk mRNA-seq of mKate2^low/GFP^high and mKate2^high/GFP^low fractions from HT29^SOX9-mKate2/KRT20-GFP dual reporter line. Two differentiation signatures are indicated in red and green; a normal and aberrant stem cell signature are indicated in orange and blue, respectively. Normalized enrichment scores (NES) and false discovery rates (FDR) for each signature are listed to the right. **c** Heatmap showing select differentiation and stem cell genes from (**b**). **d** Ranked log₂FC plot of CRISPR-Cas9 screen in HT29^SOX9-mKate2/KRT20-GFP line comparing mKate2^low/GFP^high and mKate2^high/GFP^low fractions. **e** Beta score of each gene in CRISPR-Cas9 screen targeting epigenetic regulators in HT29^SOX9-mKate2/KRT20-GFP line (3 technical replicates) comparing mKate2^low/GFP^high and mKate2^high/GFP^low fractions. **f** Distribution of sgRNA log₂FC Z-scores of top and bottom hits from CRISPR-Cas9 screen targeting epigenetic regulators. Individual sgRNAs are colored green if enriched and red if

depleted in indicated fractions of dual reporter system. **g** Overlap analysis of shared and exclusive individual sgRNA hits enriched in the 75-100% versus 0-25% GFP+ fractions from the single differentiation reporter system, as well as enriched in mKate2^low/GFP^high versus mKate2^high/GFP^low fractions or enriched in mKate2^low/GFP^high versus mKate2^low/GFP^low fractions from the dual reporter system. **h** Overlap analysis of shared and exclusive individual sgRNA hits depleted in the 75–100% mKate2 versus 0-25% mKate2+ fractions from the single stem reporter system, as well as depleted in mKate2^high/GFP^low versus mKate2^low/GFP^low fractions, or depleted in mKate2^high/GFP^low versus mKate2^low/GFP^high fractions from the dual reporter system. **i** Rank sum of each gene from the CRISPR-Cas9 screen targeting epigenetic regulators in HT29^SOX9-mKate2/KRT20-GFP line comparing mKate2^low/GFP^high to mKate2^high/GFP^low fractions. Boxplots show distribution of sgRNAs per gene. Probability density plots show distribution of enriched (green), depleted (red), and all other sgRNA (gray) rank sums (right panel).

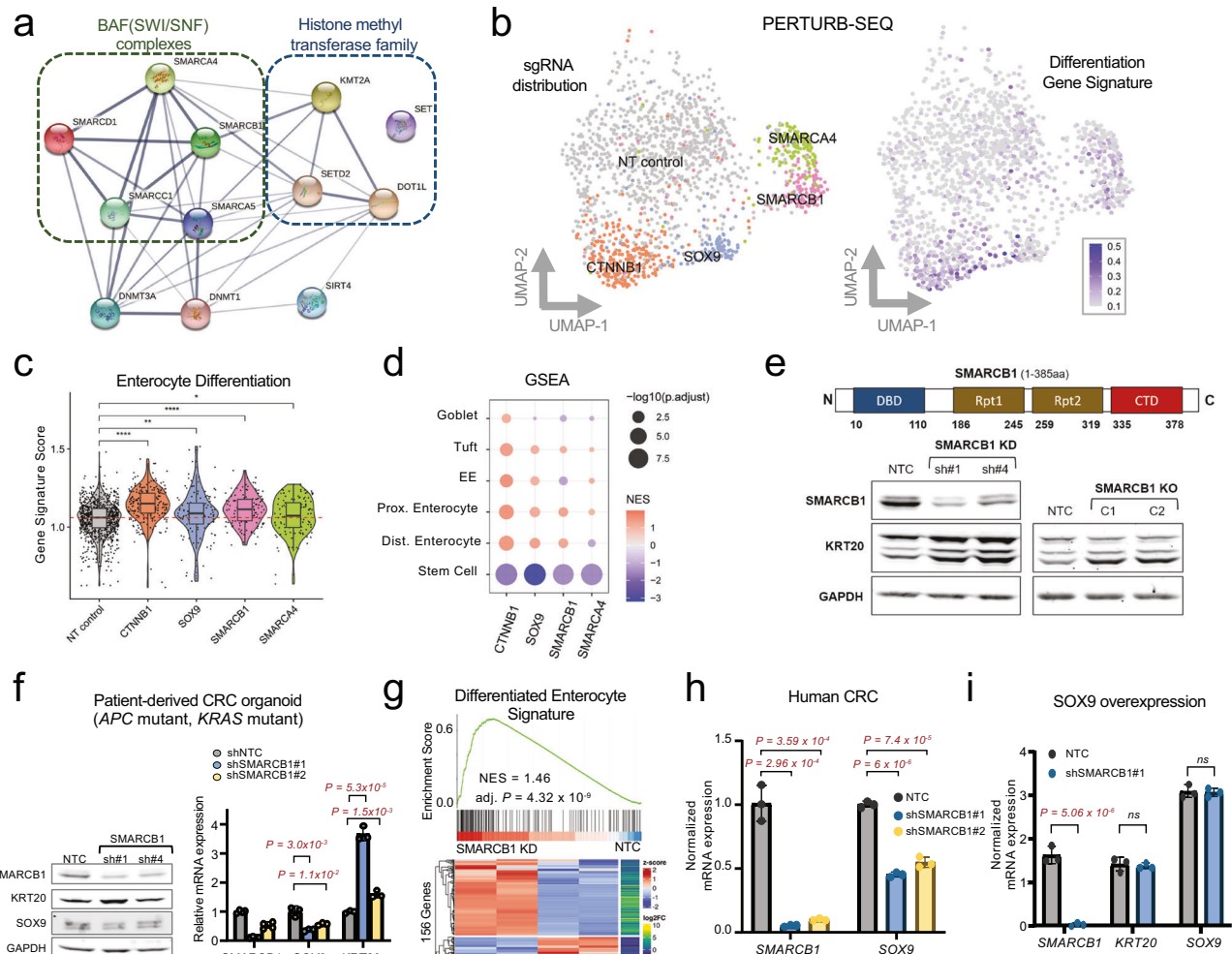

**Fig. 4 | SMARCB1 restricts differentiation in CRC models. a** Schematic of gene and protein regulatory network among genetic perturbations that promoted differentiation and impeded aberrant stem-cell activity in the dual reporter system using STRING. **b** UMAP representation of CRISPR-Cas9 perturbation of select genes coupled with scRNA-seq readout (Perturb-seq). sgRNA assignment of each cell (left). Representation of intestinal differentiation gene signature (right)[79]. **c** Violin plot depicting the enterocyte gene expression signature in cells with NT control, CTNNB1, SOX9, SMARCB1, and SMARCA4 sgRNAs. *P*-values determined by Wilcoxon test (**p* ≤ 0.05; ***p* ≤ 0.01; ****p* ≤ 0.0001) **d** GSEA of single cell transcriptomes from cells with CTNNB1, SOX9, SMARCB1, and SMARCA4 sgRNAs relative to NT controls using distinct differentiation and one stem cell signature[79]. **e** Immunoblot of SMARCB1, KRT20, and GAPDH in HT29 SMARCB1 KD cell lines using 2 shRNAs and HT29 SMARCB1 KO clones using CRISPR-Cas9; clone C1 has 32.9% and clone C2 has 72.6% editing. **f.** Immunoblot of SMARCB1 and KRT20 (left) and mRNA expression (right) of *SMARCB1*, *SOX9*, and *KRT20* in patient-derived CRC organoid expressing NTC and 2 SMARCB1 shRNAs. **g** GSEA (top) and heatmap (bottom) of differentiation gene expression signature in bulk RNA-seq data from the patient-derived CRC organoid expressing NTC and 2 SMARCB1 shRNAs. **h** mRNA expression of *SMARCB1* and *SOX9* in HT29 SMARCB1 KD cell lines using 2 shRNAs. **i** mRNA expression of *SMARCB1, KRT20, SOX9* in HT115 control and SMARCB1 KD lines engineered to conditionally overexpress SOX9. Source data are provided as a Source Data file.

perturbation corresponding to suppression of a canonical stem cell program (Fig. 4d). These results indicate that specific BAF complex components may participate in CRC pathogenesis by restricting differentiation and perturbation of these factors promote differentiation.

We next focused on SMARCB1 given its consistent and potent ability to induce differentiation upon perturbation relative to KMT2A and SMARCA4 (Fig. 4c, d and Supplementary Fig. 4a–c). Indeed, shRNA-mediated SMARCB1 knockdown (KD) and individual clones with partial SMARCB1 KO (homozygous deletion was not observed) demonstrated induction of differentiation as indicated by elevated KRT20 expression (Fig. 4e and Supplementary Note). SMARCB1 KD reduced SOX9 and induced KRT20 mRNA expression in colon premalignant organoids from a patient with familial adenomatous polyposis (FAP), albeit to a lesser extent, likely due to strong stem cell cues from WNT/R-spondin/Noggin containing conditioned media (Supplementary Fig. 4d); similar results were achieved with KMT2A KD but not SMARCA4 KD (Supplementary Fig. 4d). SMARCB1 KD in a patient-

derived CRC organoid (PDO) harboring *APC* and *KRAS* mutations reduced SOX9 and induced KRT20 mRNA and protein expression (Fig. 4f). Bulk RNA-sequencing followed by GSEA indicated that SMARCB1 KD induced a broad differentiation program (Fig. 4g); stem cell programs were not reduced, however, diverging from our cell line Perturb-seq data. Furthermore, SMARCB1 KD enhanced the ability of pro-differentiation drug all-trans retinoic acid (ATRA)[35,36] to induce *KRT20* expression in human CRC (Supplementary Fig. 4e). These results suggest that SMARCB1 restricts differentiation in different models of CRC.

To understand the functional relationship between SMARCB1 and SOX9, we started by measuring SOX9 expression levels in SMARCB1 KD lines, which showed a consistent ~50% reduction in *SOX9* mRNA transcripts in HT29, HT115, and PDO CRC models (Fig. 4f, h and Supplementary Fig. 4f). These results raised the possibility that SMARCB1 is part of the transcriptional regulation of SOX9 in CRC. Since SOX9 overexpression suppresses *KRT20* expression in CRC[24]

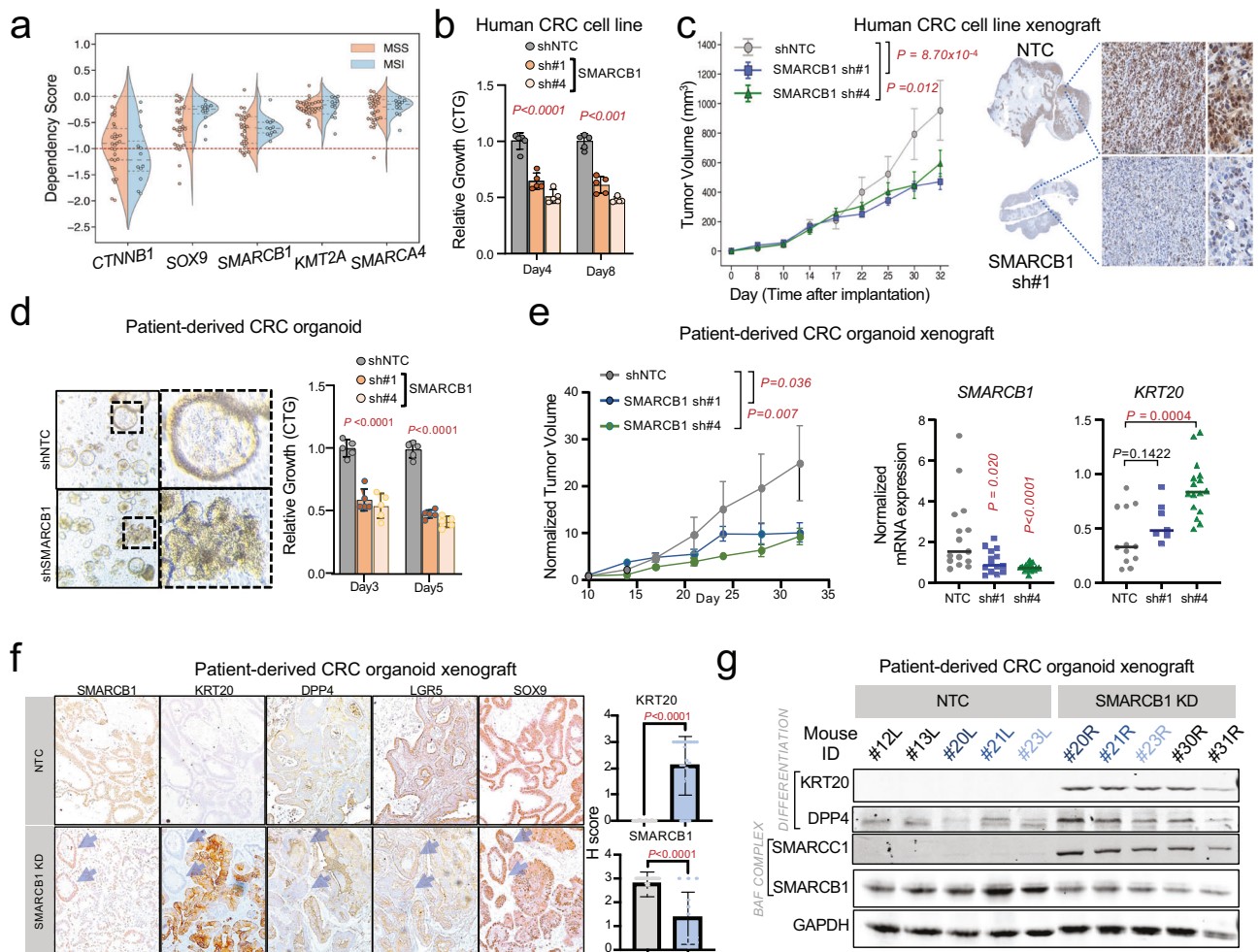

**Fig. 5 | SMARCB1 is a CRC dependency and required for in vivo tumor growth.**
**a** Distribution of CTNNB1, SOX9, SMARCB1, KMT2A, and SMARCA4 dependency scores across microsatellite stable (MSS) and microsatellite instability (MSI) CRC cell lines. Dotted red line at a dependency score −1.0. **b** Relative growth of HT29 CRC cell line expressing NTC or two shRNAs targeting SMARCB1. **c** Tumor volume of HT29 CRC cell line xenografts expressing NTC shRNA ($n = 6$), SMARCB1 sh#1 ($n = 6$), or SMARCB1 sh#4 ($n = 6$). Tumor volume data shown as mean ± SEM. SMARCB1 immunohistochemistry of representative xenografts at experiment endpoint (right). **d** Morphology and relative growth of patient-derived CRC organoid upon SMARCB1 knockdown compared to NTC. **e** Relative growth of patient-derived CRC organoid xenograft expressing NTC shRNA ($n = 7$), SMARCB1 sh#1

($n = 6$), or SMARCB1 sh#4 ($n = 8$). Relative growth data shown as mean ± SEM. Normalized mRNA expression of *SMARCB1* and *KRT20* in patient-derived CRC xenografts at experimental endpoint. **f** Immunohistochemistry of SMARCB1, KRT20, DPP4, LGR5, and SOX9 in patient-derived CRC organoid xenografts expressing NTC or SMARCB1 shRNA. Blue arrows indicate two crypts that escaped SMARCB1 KD. Quantification of SMARCB1 and KRT20 shown in the right panel. **g** Immunoblot of differentiation markers KRT20 and DPP4, BAF complex members SMARCB1 and SMARCC1, and loading control GAPDH in patient-derived CRC organoid xenografts. Xenografts marked in blue represent paired xenografts grown in the same mouse; L = left and R = right. Source data are provided as a Source Data file.

(Supplementary Fig. 4g), we asked whether SMARCB1 was required for the SOX9-mediated differentiation block. SMARCB1 KD prevented the ability of SOX9 overexpression to suppress *KRT20* (Fig. 4i and Supplementary Fig. 4g), suggesting that SMARCB1 is required for SOX9-mediated impairment of differentiation.

**SMARCB1 suppression impairs CRC growth in vitro and in vivo**
Since promoting differentiation has anti-tumor effects in CRC, we next sought to characterize if SMARCB1 is a dependency. Using Broad's Dependency Map portal[9], we analyzed the relative dependency for *CTNNB1*, *SOX9*, *SMARCB1*, *KMT2A*, and *SMARCA4* in microsatellite stable (MSS, $n = 30$) and instability (MSI, $n = 12$) CRC cell lines (Fig. 5a). Not surprisingly, *CTNNB1* and *SOX9* showed the strongest dependency across CRC cell lines, consistent with previous findings[24,27,30,37]. Among the implicated genes, *SMARCB1* showed the strongest dependency ($M = −0.643$, 95% CI = [−0.753, −0.533]) compared to *KMT2A* ($P = 0.0001$, $M = −0.225$, 95% CI = [−0.288, −0.163]) and *SMARCA4*

($P = 0.0005$, $M = -0.265$, 95% CI = [−0.358, −0.163]) (Fig. 5a), in proportion to their relative effects on differentiation (Fig. 4). While SOX9 shows a near-preferential dependency in MSS relative to MSI CRC cell lines ($t(40) = −1.814$, $p = 0.0772$), the impact of SMARCB1 loss was similar in CRC cell lines irrespective of MSI status ($t(40) = −0.656$, $p = 0.516$) (Fig. 5a). Among 30 MSS CRC cell lines, *SOX9* and *SMARCB1* showed comparable distribution of dependency strengths ($p = 0.190$, 95% CI = [−0.0764, 0.633]) (Supplementary Fig 5a). We next demonstrated that shRNA-mediated SMARCB1 KD significantly reduced in vitro proliferation (Fig. 5b) and in vivo tumor growth using a human CRC cell line xenograft (Fig. 5c and Supplementary Fig. 5b). Tumor xenografts maintained SMARCB1 KD throughout the experiment as shown by SMARCB1 IHC (Fig. 5c).

As PDOs are increasingly adopted as the favored preclinical model for their capacity to faithfully recapitulate patient response to small-molecule chemical perturbations[38,39], we next evaluated the requirement for SMARCB1 in this human CRC model (*APC*^c.835-8A>G, *KRAS*^G12V).

SMARCB1 KD reduced organoid proliferation by 50% in vitro, which was associated with morphological changes (Fig. 5d). SMARCB1 KD also impaired tumor growth in vivo using a xenograft model (Fig. 5e and Supplementary Fig. 5b). Two organoid xenografts escaped SMARCB1 KD targeted by their shRNA and showed greater tumor growth compared to controls, reinforcing selective pressure against SMARCB1 loss (Supplementary Fig. 5c). Tumor xenografts at experimental endpoint showed reduced SMARCB1 and elevated KRT20 mRNA expression by RT-PCR (Fig. 5e). Consistently, SMARCB1 KD xenografts showed reduced SMARCB1 and elevated KRT20 protein levels by histopathology and immunoblot (Fig. 5f, g, paired xenografts marked in different shades of blue). A second differentiation marker, DPP4, was also induced in SMARCB1 KD PDO xenografts (Fig. 5f, g). Notably, two crypts that escaped SMARCB1 KD did not induce KRT20 or DPP4 expression (Fig. 5f, arrows). While stem cell marker LGR5 was modestly reduced, SOX9 levels did not change in SMARCB1 KD xenografts (Fig. 5f). Interestingly, SMARCC1, a core member of all 3 BAF complexes, was only expressed in SMARCB1 KD xenografts, suggesting compensation for SMARCB1 loss (Fig. 5g). Collectively, these results demonstrate a requirement for SMARCB1 in restricting differentiation and tumor growth in CRC.

## Discussion

Genetic screens that utilized functional readouts have helped identify regulators of biological processes and pathological conditions[40]. Impaired differentiation is a hallmark of cancer[21], perhaps most recognized and well-characterized in hematological malignancies. Acute myeloid leukemia (AML) displays an arrest in differentiation, which leads to expansion of an immature and self-renewing population of neoplastic cells. Of translational importance, therapies that promote myeloid differentiation have shown great efficacy in acute promyelocytic and IDH-mutant subtypes of AML[23]. These translational successes have motivated genetic and chemical screens to identify regulators and drugs targeting differentiation blocks, respectively. By applying a chemical compound library to a conditional HoxA9-mediated differentiation block[41], inhibition of the enzyme dihydroorotate dehydrogenase was identified as potential pro-differentiation therapeutic strategy[42]. A genome-wide CRISPR screen in acute myeloid leukemia using CD14 expression as a marker of AML blasts demonstrated that ZFP36L2 promotes differentiation blockade by destabilizing myeloid maturation transcripts[43]. Similarly, using CD11b surface marker expression as a readout of myeloid differentiation, a CRISPR screen targeting chromatin remodelers and RNA binding proteins revealed that KAT6A and ENL collaborate in a pro-leukemogenicity transcription circuit that stalls differentiation[44].

Aberrant stem cell-like activity imparts differentiation defects in CRC development[24,26,28,30]. Given that unrestricted WNT activation is a critical initiating event in CRC pathogenesis, most frequently via loss-of-function APC mutations[45–47], it is natural that therapeutic discovery efforts and screening approaches have been directed towards disrupting this potent developmental pathway[27,48–50]. In DLD1 and HCT116 CRC cell lines engineered to express an exogenous β-catenin-responsive, luciferase-based reporter, CDK8 was identified as being necessary for β-catenin-driven malignancy from a focused shRNA screen[51]. With a different exogenous WNT-responsive reporter approach on the same CRC cell lines, TCF7 was identified as a positive regulator of CRC while TCF7L2 was shown to have tumor suppressor properties in CRC[17]. When DLD1 cells were engineered to express a similarly designed, exogenous β-catenin-responsive reporter, AGGF1 was found to regulate WNT/β-catenin signaling by a genome-wide siRNA screen[52]. An alternative genetic perturbation screening strategy by gene-trap retrovirus employed on a haploid CRC cell line engineered to express an exogenous Wnt-responsive, fluorescence-based reporter nominated a different set of regulators of WNT pathway in CRC[53]. Due to these broad results using exogenous reporter-based screens, efforts have

been made to engineer endogenous reporters[28] compatible with forward genetic screening. The histone modifier KMT2A was recently nominated to be a regulator of aberrant WNT/β-catenin signaling by a genome-scale CRISPR screen in DLD1 CRC cells engineered to express fluorescence-based reporter at the endogenous locus of either AXIN2 or cMYC, direct targets of β-catenin[19]. They also found that there was a disconnect between exogenous and endogenous reporter results. These screens have relied on capturing a loss of signal, which has the potential for false positives compared a gain in signal[20]. We therefore decided to manufacture a dual endogenous reporter system capable of capturing information from either pole of the stem cell-differentiation axis.

Our results indicate that utilizing the dual reporter improved signal-to-noise and consistency among replicates while reducing sgRNA variance in the CRISPR screen compared to each single reporter, providing greater confidence in nominated functional candidates. By evaluating gain and loss of signal from opposing pathological processes, we aimed to identify regulators along a specific axis and reduce the chances of false positive results. Another approach would have been to use both gain- and loss-of-function screening, which was recently shown using parallel CRISPRi and CRISPRa screens to investigate functional regulators of interferon and IL-2 signaling in T-cells[54]. The dual reporter can also facilitate identification of factors that only impact one reporter (e.g., differentiation) without influencing the other (e.g., stem cell activity). This platform can be scaled up to perform genome-wide CRISPR screens and is also suitable for small molecule chemical screens, which can be combined with high content imaging to accurately measure phenotypic readouts of stem cell and differentiation activity.

We focused our validation studies on three genes - KMT2A, SMARCA4, and SMARCB1 - that are part of two functional modules (Fig. 4a). KMT2A (lysine methyltransferase 2A; also known as mixed lineage leukemia 1 or MLL1) is a methyl transferase that is characterized for methylating lysine 4 on histone 3 (H3K4), which regulates enhancers that function in early development and hematopoiesis[55]. Our data indicated that KMT2A negatively regulates differentiation in the colon. Disruption of KMT2A in zebrafish resulted in premature differentiation of neurons and decrease proliferation[56], suggesting that it may restrict differentiation across multiple tissue. With respective to cancer biology, KMT2A/MLL1 is altered by chromosomal rearrangement in poor prognosis leukemias[57]. In a mouse model of colon cancer, Kmt2a/Mll1 was required for WNT-driven adenomas by sustaining activating H3K4 tri-methylation at stem cell genes, including Gata4/6 transcription factors[58]. Consistently, a genome-scale CRISPR screen using an endogenous WNT reporter identified KMT2A as a regulator of canonical β-catenin transcriptional activity, and the loss of KMT2A by CRISPR-mediated knockout led to induction of KRT20 and suppression of intestinal stem cell markers (LGR5 and ASCL2) in CRC cells[19]. These studies suggest that suppression of KMT2A leads to intestinal differentiation by disrupting β-catenin/WNT signaling in our CRC models.

SMARCA4 and SMARCB1 both encode DNA-binding subunits of the chromatin remodeling BAF complex[59]. SMARCA4 encodes BRG1, which functions as an ATPase in the BAF complex. High expression of SMARCA4 is observed in most cancers and correlates with poorer survival in a subset[60]. In a model of colon cancer, SMARCA4 and PRMT1 were shown to promote CRC progression cooperatively via EGFR signaling[61]. There are multiple drugs inhibiting SMARCA4 under preclinical evaluation[62,63]. Most recently, a SMARCA2/4 degrader disrupted promoter and enhancer interactions and suppressed tumor growth in prostate xenograft models[64].

SMARCB1, which encodes SNF5, is well-known for its tumor suppressor function across multiple cancers including pediatric CNS cancers and sarcomas[65,66]. Embryonic rhabdoid tumors harbor homozygous SMARCB1 mutations[67]; SMARCB1 binds to and maintains enhancers involved in differentiation in these tumors[68]. Indeed,

SMARCB1 engages lineage-specific enhancers to maintain developmental and differentiation programs[69]. Gene expression profiling of induced pluripotent stem cells and neural progenitors revealed that the transcriptional effect of SMARCB1 is context dependent[70]. Mutations in SMARCB1 also lead to neurodevelopmental diseases[71] and suppression of BAF complex components promote premature neuronal differentiation[72]. SMARCB1 loss led to induction of differentiation in our CRC models, which may be due to enhancer specificity in colon epithelium[68]; the underlying molecular mechanism, though, requires further elucidation. SMARCB1 loss has also been shown to induce β-catenin activity[73], which would promote stem cell behavior in colonic tissue and contradict our model. While a SMARCB1 inhibitor does not exist, SMARCB1-deficient tumors appear to be co-dependent on EZH2, which has an oral-form small molecule inhibitor. EZH2 inhibitors induced differentiation and apoptosis in SMARCB1-deleted malignant rhabdoid tumor xenograft[66,74]. If SMARCB1 inhibitors are developed, their therapeutical potential in CRC may be realized in combination with EZH2 inhibitors.

## Study limitations

The dual reporter relies on SOX9 and KRT20 as markers for aberrant stem cell-like and differentiation activity. While these are faithful markers, they may not capture the full complexity of these transcriptional programs. Perturbation of epigenetic regulators did not induce differentiation to the same magnitude as SOX9 suppression. This may reflect that many of these factors function in complexes and their loss may lead a partial defect in complex activity. The impact of losing these factors on differentiation should also be examined using dynamic assays in addition to single timepoint experiments. The molecular mechanism(s) by which SMARCB1 restricts differentiation in CRC has not been addressed by this study and requires further investigation.

# Methods

## Ethical statement

Protocols were approved by the Internal Review Board of the Dana Farber Cancer Institute, Boston, Massachusetts, USA (protocols 13-189 and 14-408). Written consent was obtained from all participating patients, which included consent to publish results. All mice and experimental protocols were approved by Institutional Animal Care and Use Committee (IACUC) of Dana-Farber Cancer Institute (11-009). The maximal tumor size permitted by the IACUC is 2 cm[3], which was not exceeded.

## Cell culture

All cell lines were maintained at 37 °C with 5% $CO_2$. The human colorectal cancer cell lines (HT115, HT29 (HTB-38) and LS180 (CL-187)) and HEK293T were obtained from CCLE core facility at the Broad Institute, MIT and used at early passage for the experiments. Cells were maintained in DMEM medium supplemented with 10% FBS and 1% penicillin/streptomycin. Mycoplasma testing was performed every 3 months and found to be negative on each check.

## Engineering endogenous stem cell and differentiation reporter system

We knocked-in mKate2 (mK2, a next-generation RFP probe) and GFP fluorescent markers in-frame at the end of the SOX9 and KRT20 coding regions using a combination of CRISPR/Cas9 and template-based homologous recombination to establish single and dual endogenous stem cell and differentiation reporter lines according to the detailed protocols (PCR template-mediated HDR, RNP complex assembly and transient transfection) below.

We designed and tested sgRNAs to target the last exon of *SOX9* in closest proximity to the stop codon. Cas9 pre-preloaded in vitro with the best performing sgRNA (see methods for T7 assay) was introduced into CRC cells for precise and efficient genome editing by electroporation. A double stranded break (DSB) followed by homology directed repair (HDR) facilitated integration of the GFP fluorescent reporter cassette.

**PCR template-mediated HDR.** Fluorophore reporter genes and antibiotic selection markers were amplified by universal primers flanked with SOX9 and KRT20 specific flaking primers to ensure the site-specific integration of the PCR products. Primer sequences are listed in Supplementary Data 1.

## RNP complex assembly

AltR RNA oligos were ordered from IDT. After reconstitution (100 µM) tracrRNA and crRNA were mixed equimolarly and incubated for 5 min at 95 C. After denaturation, products were incubated for 5 min at room temperature for complex formation.

**Transfection and KI cell line preparation.** HT29 cells were plated two days before transfection, to reach 70–80% confluency at the time of transfection. $0.5 \times 10^6$ cells per transfection were collected for each condition. Cells were transfected with RNP complex and HDR templates by nucleofection with SF Cell Line 4D-Nucleofector X Kit (Lonza) using 20-ml Nucleocuvette Strips, as described by the manufacturer (Program FF137). Cells were immediately resuspended in 100 ul of culturing medium and plated into 1.5 ml of pre-warmed culturing medium in 24-well tissue culture plates. T7E1 assay, and site-specific PCR reactions were performed 72 h after nucleofection in order to check cleavage efficiency and integration, respectively. After confirming the correct integration and one-week regeneration cells were plated into 15 cm dish. The next day media was changed to antibiotic containing media (500 µg/ml neomycin and 20 µg/µl blasticidin) and antibiotic selection was performed. After an additional week, formed colonies were collected by trypsinization and resuspended in culture media. After regeneration culturing period (3–5 days), cell lines were tested by FACS analysis and site-specific PCR.

## Composition of control sgRNA and shRNA library

Six sgRNAs/shRNAs per gene targeting GFP, mKate2, SOX9, KRT20 and sgRNAs/shRNAs targeting other genes including control genes in a total of 76 sgRNAs and 154 shRNAs (Supplementary Data 3).

## Design of focused epigenetic sgRNA library

For focused epigenetic regulator screens, we created a small library from selected members of epigenetic regulator families (Histone Acetylation (BRD), Histone deacetylation (HDAC & SIRT), Histone Lysine Methyltransferase (SET & KMT), Chromatin remodeling BAF(SWI/SNF) complex, and DNA Methyltransferases (DNMT)) for a total of 78 genes. Four hundred sixty-six sgRNAs were selected from the H3 knockout library (Addgene # 133914) from Brown/Liu labs.

Regarding the control sgRNAs, we used SOX9, KRT20, GFP, mKate2, PROM1, LRIG1 as positive controls and non-targeting sgRNAs as negative controls. sgRNAs were designed using the Chop-Chop algorithm[75]. The total of 542 sgRNA were included in the library (Supplementary Data 2). Its abundance was determined by deep amplicon sequencing.

## Oligo design, cloning and library preparation

The pCC_01 - hU6-BsmBI-sgRNA(E + F)-barcode-EFS-Cas9-NLS-2A-Puro-WPRE plasmid (Addgene #139086) was used for CRISPR screen and validation. Briefly the plasmid was digested with Esp3I (NEB#R0734) at 37 °C overnight. The 13 kb band was purified with gel purification kit (Takara Nucleospin® Gel and PCR Clean-up Midi). Eighty-two base pair single-stranded oligos were designed according to the following structure:

AGGCACTTGCTCGTACGACGCGTCTCGCACC - gNNNNNNNNNN NNNNNNNNN -GTTTAGAGACGTTAAGGTGCCGGGCCCACAT

Thirty-one base pair 5′ constant sequence compatible with the linearized (*Esp3I*) pCC_01 vector (3′ end of the U6 promoter) followed by the 19 bp sgRNA sequence adding an extra g base for the proper sgRNA transcription, followed by the 3′ vector compatible (5′ end of the sgRNA scaffold) sequence. Single stranded oligo pools were ordered from IDT (oPools™ Oligo Pools) and built into the digested PCC_01 vector using NEBuilder HiFi DNA Assembly Master Mix (E2621L, New England Biolabs) according to the manufacturer's instruction. The list of all oligos can be found in the Supplementary Data 3. Assembled gRNA libraries were purified by AMPure XP magnetic beads (Beckman Coulter).

## Flow cytometry and cell sorting (FACS)

FACS was performed on the FACSAria III platform (BD) at DFCI Flow-core facility. For GFP labeled cell collection, 488 nm laser and 585/42 filter were used. For mKate2 651 nm laser and 610/20 filter were used. For single channels, 4 fractions were collected based on histogram profile, using 4-way sorting, approximately 25% per each of under curve area. We named these fractions: low (0–25%), mid low (25–50%), mid high (50–75%), high (75–100%). For dual reporter, 4 fractions were collected, 2.5% of each, including dual negative, dual positive, GFP positive and mKate2 positive fractions. Cells were collected into 5-ml tubes. After collecting cell pellet, direct PCR was performed using Phire Tissue master mix (F170L, Thermo), according to the manufacturer recommendations. Guide RNA abundance was determined from each fraction by deep amplicon sequencing. Library preparation was performed according to the Broad Institute recommended protocol at the GPP portal using Argon primers for the demultiplexing. (https://portals.broadinstitute.org/gpp/public/resources/protocols).

## Individual gRNA cloning for validation

All the sequences associated with the validation are listed in the Supplementary Data 4.

For the validation, two individual oligos, top and bottom were ordered from IDT as designed.

Top strand: protospacer sequence 5′-CACCGNNNNNNNNNNNNNN NNNNNNN-3′

Bottom strand: reverse complement 5′-AAACNNNNNNNNNNNNNN NNNNNNNNC-3′

The oligos were phosphorylated with T4 PNK enzyme (NEB #M0201) and annealed after denaturation. The digested plasmid and phosphorylated oligo cassettes were ligated with T4 DNA ligase (NEB #M0202L) at room temperature for 10 minutes. The ligated vectors were introduced by heat-shock transformation into NEB Stable (NEB #C3040H) chemically competent *Escherichia coli* and propagated on selective agar plates. Colonies were picked and the protospacer sequences were confirmed with Sanger sequencing (Genewiz). Plasmid DNAs were extracted with QIAGEN Plasmid Plus MIDI kit (QIAGEN 12941) for downstream applications.

**pLKO.1-TRC plasmid.** (Addgene #10878) was used for shRNA constitutive knockdown in cancer cells. The plasmid was digested with AgeI (NEB #R3552) and EcoRI (NEB #R3101L) at 37 °C overnight. The 7 kb band was purified as above. The single-stranded pool oligos were order from IDT.

5′-CTTTATATATCTTGTGGAAAGGACGAAACA CCGG NNNNNNN NNNNNNNNNNNNNNNN CTCGAG nnnnnnnnnnnnnnnnnnn TTTTT A ATTCTCGACCTCGAGACAAATGGCAGTATT-3′

The digested plasmid and pool oligos were ligated with Gibson reaction (NEBuilder HiFi DNA Assembly Master Mix E2621) at 50 °C for 1 h and purified with 1.0× AmPure Bead (AmPure XP A63881) before the

transformation. Each bacterial colony was picked and Sanger sequenced before proceeding to plasmid propagation.

**pRSITEP-U6Tet-sh-noHTS-EF1-TetPep-2A-Puro plasmid.** (Cellecta #SVSHU6TEP-L-CT) was used for shRNA inducible knockdown in colon organoids. The plasmid was digested with BbsI (NEB R3539L) at 37 °C overnight and purified with 1.0× AmPure Bead. Two individual oligos, top and bottom were ordered from IDT as designed.

Top strand: hairpin sequence 5′-G NNNNNNNNNNNNNNNNNNNNN CTCGAG nnnnnnnnnnnnnnnnnnnnn TTTT-3′

Bottom strand: reverse complement 5′-CGAA AAAA NNNNNNNN NNNNNNNNNNNNN CTCGAG nnnnnnnnnnnnnnnnnnnnn C-3′

The oligos were phosphorylated and annealed with T4 PNK enzyme. The digested plasmid and phosphorylated oligos were ligated with T4 DNA ligase at room temperature for 10 min. The ligated vectors were heat-shock transformed to NEB Stable competent *E. coli*. The plasmids were extracted with QIAGEN Plasmid Plus MIDI kit and the hairpin sequences were confirmed with Sanger sequencing.

**pLIX403-Blasticidin.** (Addgene #158560) was used for doxycycline-inducible SOX9 overexpression. In brief, SOX9 full length and GFP open-reading frames contained in donor plasmids were swapped with CcdB toxin using Gateway cloning as described in the paper[24].

## Lentivirus transfection for validation

To generate lentiviruses for validation experiments, HEK293T cells were co-transfected with sgRNA expression vectors and lentiviral packaging constructs psPAX2 and pMD2.G (VSV-G) in a 2:1:1 ratio using X-tremeGENE 9 DNA Transfection Reagent (Roche) according to the manufacturer's instructions. Cell culture media was changed the following day and lentiviral supernatant was harvested 24 h and 48 h later and filtered through a 0.22 μm filter (Celltreat). Lentiviruses were aliquoted and stored at −80 °C until use.

To perform lentiviral infection, the CRC cells were plated in a 6-well flat bottom plate and infected with 0.5 mL virus in media containing 8 mg/mL polybrene overnight. The media was changed the next day. After 2 days, puromycin or blasticidin were started at 5–8 μg/mL and 20 μg/mL respectively and continued until the negative control cells died.

## CRISPR knockout single clone derivation

HT29 cell line with SMARCB1 CRISPR knockdown was trypsinized and filtered through 20 μm then 10 μm filter in order to make single cells. Single cells were counted and plated at 1000 cells in 10 cm culture dish. After 1–2 weeks, multiple clones were picked and collected DNA to confirm genomic sequence around the cut site.

## ATRA treatment

HT29 was treated with all-trans retinoic acid (ATRA) at 10 μM concentration for 48 h.

## Immunoblot analysis

Cells were lysed in RIPA buffer supplemented with a protease inhibitor cocktail (Roche) and phosphatase inhibitor (Cell Signaling 5870s). Protein were measured with Pierce™ Rapid Gold BCA Protein Assay Kit (Thermo Fisher A53227) and denatured in 4× Laemmli buffer (Biorad 1610747). Whole cell extracts were resolved by 8–16% Tris-glycine polyacrylamide gel (Invitrogen XP08165BOX), transferred to PVDF membranes with iBlot™ 2 transfer device (Thermo Fisher IB21001) using P0 protocol, and probed with indicated primary antibodies. Bound antibodies were detected with LI-COR IRDye ® 680/800CW anti-rabbit/mouse secondary antibodies. Primary antibodies listed in the Supplementary Data 5.

## RNA isolation, RT-qPCR

Total RNA was isolated using the RNeasy Mini Kit (Qiagen 74004) and cDNA was synthesized using the iScriptTM Reverse Transcription Supermix for RT-qPCR (Bio-Rad 1708840). Gene-specific primers for SYBR Green real-time PCR were synthesized by Integrated DNA Technologies. RT-qPCR was performed and analyzed using CFX96 Real-Time PCR Detection System (Bio-Rad 1845096) and using Power SYBR Green PCR Master Mix (Thermo Fisher 4368577). Relative mRNA expression was determined by normalizing to ACTB expression.

## Pooled CRISPR screen analysis

Abundance of sgRNA reads was counted using a custom Python-based adaptation of the Aho-Corasick string-search algorithm to (1) demultiplex FASTQ files and assign each read to a known cell fraction by searching for the associated barcode in each read and to (2) read count the number of sgRNA in each cell fraction by searching for the sgRNA sequence in each read without allowing for any mismatches. To process and analyze CRISPR screen data, the following softwares, packages and modules were used: Python v.3.9.13 (Pandas v.1.5.3, Numpy v.1.26.1, Seaborn v.0.11.2, Scipy v.1.11.3, Matplotlib v.3.7.2, Statsmodels v.0.13.2, GSEApy v.1.1.1), MaGeCK v.0.5, STRING v.12.0.

## Normalization of sgRNA representation

First, total read count normalization assumes that the total expression of sgRNA reads is the same under the different experimental conditions. The total read count normalization step divides the read count for each sgRNA by the total number of read counts across all sgRNA in the sorted cell fraction. After dividing by total sample size, the output normalized read counts represent the proportion of each sgRNA read's abundance relative to the total sgRNA read count Second, library pool normalization considers the uneven sgRNA representation within the library and assumes that each sgRNA read within the pool should have similar read counts on average in each sorted cell fraction. The library pool normalization step divides the read count for each sgRNA in every sample by the mean of the read counts for each sgRNA from 3 independent library pool control populations. Third, population normalization assumes that most sgRNA should have similar read counts on average between different sorted cell fractions from the same cell population. The population normalization step divides the read count for each sgRNA in each fraction by the total read count for each gRNA across all fractions of the cell population. Normalized sgRNA read counts are compared between different cell fractions sorted based on fluorescence intensity using the single and dual endogenous reporter systems.

## Flow sorting using single endogenous reporter system

Cell populations with the knock-in single endogenous reporter system were sorted based on fluorescence intensity into four equal fractions (0–25%, 25–50%, 50–75%, and 75–100% fluorescence intensity) to represent the gradual progression of cell state differentiation (using GFP) or the stem cell program (using mKate2) in colorectal cancer cell lines. Three informative pairwise comparisons of sgRNA read count abundance to the cell fraction with the lowest fluorescence intensity range (0–25%) were made: 25-50% vs. 0–25%, 50–75% vs. 0–25%, and 75–100% vs. 0–25%.

## Flow sorting using dual endogenous reporter system

Four fractions for each sample were obtained using flow sorting based on fluorescent intensity of green fluorescent protein and red fluorescent protein using the dual endogenous reporter system. For cell lines with knock-in of the dual reporter system, the mKate2$^{low}$/GFP$^{high}$ fraction represents cells within the population which exhibit relatively high green fluorescent intensity and the mKate2$^{high}$/GFP$^{low}$ fraction represents cells within the population which exhibit relatively high red fluorescent intensity. The mKate2$^{low}$/GFP$^{low}$ fraction and mKate2$^{high}$/

GFP$^{high}$ fraction represents cells within the population which exhibit relatively low and high green and red fluorescent intensity respectively. Between the four sorted cell fractions, five possible informative pairwise comparisons can be made to assess the differential fold change of normalized sgRNA abundance in different cell states: mKate2$^{low}$/GFP$^{high}$ vs. mKate2$^{high}$/GFP$^{low}$, mKate2$^{low}$/GFP$^{high}$ vs. mKate2$^{low}$/GFP$^{low}$, mKate2$^{high}$/GFP$^{low}$ vs. mKate2$^{low}$/GFP$^{low}$, mKate2$^{low}$/GFP$^{high}$ vs. mKate2$^{high}$/GFP$^{high}$, and mKate2$^{high}$/GFP$^{low}$ vs. mKate2$^{high}$/GFP$^{high}$.

## Differential sgRNA abundance analysis

Log$_2$ fold change (log$_2$FC) of normalized sgRNA abundance and Z-score of normalized sgRNA abundance was used to compare differences in sgRNA abundance between different sorted cell fractions. The Model-based Analysis of Genome-wide CRISPR/Cas9 Knockout (MaGeCK) tool with default parameters was used to test for differential abundance of normalized sgRNA read counts between different sorted fractions of multiple technical replicates and estimate the beta score, a measure of gene essentiality, using MaGeCK Maximum Likelihood Estimation (MLE). Beta scores and Benjamini-Hochberg-adjusted $P$-values generated from differential sgRNA abundance between two different sorted cell fractions represent the extent of positive or negative gene selection across multiple technical replicates.

## Identifying enriched/depleted sgRNA in CRISPR screens using the rank sum scoring method

Three technical replicates of the focused epigenetic screen using the dual endogenous reporter system in the HT29$^{SOX9-mKate2/KRT20}$-GFP genome-edited cell line were conducted. Three technical replicates of the focused epigenetic screen using the single endogenous reporter system in the HT29$^{SOX9-mKate2}$ and HT29$^{KRT20-EGFP}$ genome-edited cell line were also conducted. Comparing the mKate2$^{low}$/GFP$^{high}$ fraction and mKate2$^{high}$/GFP$^{low}$ fractions in the dual reporter system was done since this comparison yielded the largest range of log$_2$ fold change of normalized sgRNA abundance. Comparing the fraction with the highest fluorescence intensity range (75-100%) and the fraction with the lowest fluorescence intensity range (0–25%) in the single-reporter system was done since this comparison yielded the largest range of log$_2$ fold change of normalized sgRNA abundance.

To identify consistently enriched and depleted sgRNA hits in the focused epigenetic library across different replicates with different effect sizes, a rank sum method was used. First, the log$_2$ fold change of normalized read count abundance between two fractions was determined for each sgRNA in each replicate. Second, within each replicate, the sgRNA were rank ordered from highest fold change to lowest fold change, such that sgRNA assigned a lower rank value (closer to 0) had a higher fold change value. Third, the ranks of each sgRNA were summed across each replicate to generate the rank sum score of each sgRNA. For sgRNAs targeting a gene to be considered enriched or depleted using their assigned rank sum scores, there must be at least 2 sgRNAs targeting the same gene that are within the top or bottom 15% of all sgRNA rank sum scores respectively. This rule was used to exclude single outlier sgRNA hits that were enriched or depleted despite other sgRNA targeting the same gene not being enriched or depleted.

## PERTURB-seq

pCC_01 and pCC_09 (Addgene #139094) were modified to incorporate Capture Sequences 1 and 2 at the 3' end of guide sequences. Oligo cloning and library preparation were done as described above. The Capture Sequences allowed sgRNAs to be captured within the same cell when cells were processed according to the Chromium Single Cell 3' Reagent Kits v3.1 with Feature Barcoding technology for CRISPR screening protocol (10X Genomics, CG000205). For analysis,

FASTQ files of GEX and CRISPR libraries were processed with default parameters on 10X Genomics *Cell Ranger* pipeline (v7.0.0) using "GRCh38-2020-A [https://cf.10xgenomics.com/supp/cell-exp/refdata-gex-GRCh38-2020-A.tar.gz]" reference. Processed data were then analyzed with *Seurat* (v4.1.1)[76] and its extension *Mixscape*[77] under "R" (version 4.2.2) environment.

## Depmap gene dependency analysis

Depmap is a cancer dependency map that systematically identified known genetic dependencies using genome-scale CRISPR knockout screens of cancer cell lines in the Cancer Cell Line Encyclopedia (CCLE). For each cell line in the CCLE, genes annotated with dependency scores below −1.0 are considered essential. To characterize the dependency of validated screen hits along with known dependencies involved in the initiation and progression of colorectal cancer (CTNNB1, SOX9), we compared the dependency score distribution of the validated gene hits in the 31 microsatellite stable (MSS) and 11 microsatellite instable (MSI) colorectal cancer cell lines of the CCLE with available dependency score data.

## Patient-derived CRC organoid model

A tumor specimen from an unidentifiable patient with CRC was collected post colectomy under approval (protocol 13-189 and 14-408) by the Internal Review Board of the Dana Farber Cancer Institute, Boston, Massachusetts, USA.

Tumor tissue was treated with EDTA and then resuspended in 30–50 μl of Matrigel (BD Bioscience) and plated in 24-well plates. WNT/R-spondin/Noggin (WRN) containing DMEM/F12 with HEPES (Sigma-Aldrich) containing 20% FBS, 1% penicillin/streptomycin and 50 ng/ml recombinant mouse EGF (Life Technologies) was used for culturing colon organoids. For the first 2–3 days after seeding, the media was also supplemented with 10 mM ROCK inhibitor Y-27632 (Sigma Aldrich) and 10 mM SB431542 (Sigma Aldrich), an inhibitor for the transforming growth factor (TGF)-β type I receptor to avoid anoikis. For passage, colon organoids were dispersed by trypsin-EDTA and transferred to fresh Matrigel. Passage was performed every 3-4 days with a 1:3–1:5 split ratio. For human colon organoid culture, the previous media was supplemented with antibiotics 100 μg/ml Primocin (Invivogen), 100 μg/ml Normocin (Invivogen); serum-free supplements 1× B27 (Thermo Fisher (Gibco)), 1X N2 (Thermo Fisher (Gibco)); chemical supplements 10 mM Nicotinamide (Sigma), 500 mM N-acetylcysteine (Sigma), hormone 50 mM [Leu15]-Gastrin (Sigma), growth factor 100 μg/ml FGF10 (recombinant human) (Thermo Fisher) and 500 nM A-83-01 (Sigma), which is an inhibitor of the TGF-β Receptors ALK4, 5, and 7.

Notable genomic alterations in the patient-derived CRC organoid includes APC: c.835-8A>G (pathogenic intronic splice variant, rs1064793022) and KRAS: G12V. All genetically manipulated colon organoid lines were generated using the protocol described here[5]. shRNAs against *SMARCB1* were cloned into PLKO.1 vector. To generate lentiviruses, expression vectors were co-transfected into HEK293T cells with the lentiviral packaging constructs psPAX2 and pMD2.G (VSV-G) in a 1:1:1 ratio using X-tremeGENE9 DNA Transfection Reagent (Roche) according to the manufacturer's instructions. Cell culture media was changed the following day and lentiviral supernatant was harvested 48 h and 72 h later and filtered through a 0.45 μm filter (Millipore). Lentiviruses were aliquoted and stored at −80 °C until use.

To transduce human CRC organoids, spheroids in one well (6-well plate) were trypsinized and used for each infection. Cells were resuspended in 500 μl lentiviral supernatant with 8 μg/mL polybrene and 10 mM Y-27632, centrifuged at $600 \times g$ 37 °C 1 h, and incubated for 6 h in cell culture incubator. The infected cells were suspended in 30–50 μl of Matrigel and cultured with Wnt/R-spondin-deprived medium containing 10 mM Y-27632 and 10 mM SB431542. Colon organoids were

selected with 3 μg/ml puromycin at 24 h post infection. Xenograft experiments were performed following standard procedures involving twelve 6-week-old male, athymic nude NOD-SCID mice (homozygous for *Foxn1ⁿᵘ*).

## CRC cell line xenograft in vivo tumor model

Related animal procedures were conducted at Kineto Lab Ltd (Budapest, Hungary) in compliance with national Hungarian legislation. Experiments were performed according to FELASA recommendations for animal use under license number PE/EA/401-7/2020. Twelve female NOD-SCID mice approximately 8 weeks old (Charles River Laboratories) were used for the in vivo xenograft experiment. HT29 human colon tumor cell line derivatives (HT29 non-targeting control (NTC), HT29 SMARCB1_shRNA1# (sh#1) and HT29_SMARCB1_shRNA#4 (sh#4)) were cultured in RPMI-1640 medium with 10% fetal bovine serum (FBS) at 37 °C with 5% $CO_2$. Before injection cells were trypsinized and washed two times with FBS free medium. One and half million cells per injection was recovered in 50 μl final volume and mixed the same volume of Matrigel. Hundred microliters cell suspension were injected into each flanks subcutaneously on both side. Tumor volumes were obtained on every 3–4 days using handheld calipers to measure tumor length, width, and height to calculate cubic millimeters; mice were weighed using a digital balance. Tumors were collected at the end of the study and placed in neutral buffered formalin (NBF) for fixation. Comparison of tumor volume over the course of treatment between either SMARCB1 knockdown or non-targeting control was conducted using two-way ANOVA.

## Histopathology

Paraffin-embedded xenograft tumors were serially sectioned and mounted on microscopic glass slides. For morphological analysis, sections were serially dehydrated in xylene and ethanol, stained with H&E for histological assessment. Slides were digitized with a Panoramic Midi slide scanner using a 20× objective (3D Histech, Budapest, Hungary).

## Statistics and reproducibility

Experiments were performed in triplicate. Data are represented as mean ± s.d. unless indicated otherwise. For each experiment, either independent biological or technical replicates were conducted as noted in the figure legends and were repeated with similar results. Statistical analysis was performed using Microsoft Office or Prism 8.0 (GraphPad) statistical tools. Pairwise comparisons between group means were performed using an unpaired two-tailed Student's *t*-test or Kruskal–Wallis test as appropriate unless otherwise indicated. Multi-group comparison of means was performed using one-way ANOVA with post-hoc Tukey's HSD test as appropriate unless otherwise indicated. For all experiments, the variance between comparison groups was found to be equivalent.

## Statement of inclusion and ethics

While this research was conducted independently, without direct collaboration with local researchers, efforts were made to ensure the relevance of the study to the local, regional, and national context with respect to colorectal cancer.

The study design, implementation, and reporting adhere to the principles outlined in the Global Code of Conduct for Research in Resource-Poor Settings to the best of our ability, with careful consideration given to ethical standards and reporting transparency.

Furthermore, this research does not pose any risk of stigmatization, incrimination, discrimination, or personal harm to any researchers. Measures have been implemented to ensure the safety and well-being of all involved parties.

We acknowledge the importance of promoting equity in research collaborations and will continue to strive for inclusivity and ethical conduct in all future research endeavors.

## Reporting summary

Further information on research design is available in the Nature Portfolio Reporting Summary linked to this article.

## Data availability

Datasets generated in this study have been deposited in the Gene Expression Omnibus (GEO) database under accession code "GSE236257". Sequencing reads are aligned to the human genome assembly "GRCh38.p14 [https://www.ncbi.nlm.nih.gov/datasets/genome/GCF_000001405.40/]". Source data are provided with this paper.

## Code availability

The CRISPR screen analysis code generated in this study are available on Github (https://github.com/davidchen0420/Endogenous_Reporter) and have been deposited to Zenodo (https://doi.org/10.5281/zenodo.10658568)[78].

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

## Acknowledgements

We thank Jacob Stewart-Ornstein for generously sharing plasmids from which PCR-amplified HDR templates were utilized. We thank Pranshu Sahgal in addition to other Sethi lab members for thoughtful suggestions. Harvard Digestive Disease Center and NIH grant P30DK034854 for core services, resources, technology, and expertise. Dana-Farber/Harvard Cancer Center is supported in part by an NCI Cancer Center Support Grant # NIH 5 P30 CA06516. S.S. was supported by Bolyai (HAS) award and received funding from National Research Development and Innovation Office Hungary (FK142835). P.L. is a recipient of the Prince Mahidol Award Youth Program, Thailand. This work was funded by the Karen Grunebaum Award and Virtual Scholar Award from the Department of Defense (CA201084) to N.S.S and philanthropic support from the Jimmy Fund Walk (Opiela Family), Craig Baskin, and Howard & Wendy Cox to the Sethi Lab.

## Author contributions

S.S. designed the study, performed experiments, and revised the manuscript. D.C. performed computational analyses, developed methodology for CRISPR screening data, wrote/revised manuscript. P.L. designed and performed validation experiments and wrote/revised manuscript. P.D. performed experimental validation, wrote/revised manuscript. Z.L. analyzed Perturb-seq experiment. P.B., L.V., V.T. performed in vivo experiments. P.S. and M.G. provided the human CRC

organoid. B.W. provided funding and revised the manuscript. J.Q. provided resources and revised the manuscript. N.S.S. designed and supervised the study, obtained funding, analyzed data, assembled manuscript, and wrote/revised the manuscript.

## Competing interests

S.S. and N.S.S. are co-inventors on an unpublished PCT patent application (63/208,313) involving part of this work filed on June 8, 2022. N.S.S. is on the scientific advisory board for Astrin Biosciences and a consultant for Dewpoint Therapeutics. The remaining authors declare no competing interests.
