## [Peer Review File · Nature Communications]

REVIEWER COMMENTS

Reviewer #1 (Remarks to the Author):

In this manuscript, Sandor Spisak and colleagues utilized single and dual fluorescent reporters of endogenous genes to investigate CRC stemness and differentiation phenotypes. They performed CRISPR screening using an epigenetic focused library and identified multiple stem-regulating candidates. Perturb-seq validation indicated that SMARCB1, a subunit of the SWI/SNF BAF complex, is a negative regulator of CRC differentiation. Finally, using Broad's Depmap portal and xenograft models, they revealed that SMARCB1 is required for the growth of CRC tumors. Interference of SMARCB1 caused upregulation of KRT20 at both mRNA and protein levels, suggesting a differentiation phenotype in these xenografts.

While the authors provide a sophisticated dual reporter system in combination with epigenetic-focused CRISPR library screening to dissect CRC stemness regulation, the utilization of dual reporter to monitor CRC stem and differentiated cells itself is not novel (Mariko Shimokawa et al. Nature. 2017 May 11;545(7653):187-192). Moreover, the fidelity of reporter system as the indicator of CRC stem cells should be validated using more intensive molecular and phenotype characterization, before conducting CRISPR library screening. The validation of SMARCB1 as a CRC differentiation factor lack mechanistic insights. The molecular events leading to CRC differentiation and SOX9/KRT20 expression alterations need to be clarified. In general, while this study provides some interesting methodological and mechanistic insights into CRC stemness regulation, its novelty is hampered by insufficient molecular characterization and mechanistic depth. The manuscript may be considered for publication pending the satisfaction of following comments:

1. A drawback of this study is the use of CRC cell lines as models of CRC stemness. Monolayer cells lack the microenvironment of stem cell niche and thus may exhibit distinct feature with cancer stem cells in vivo. Because of this difference, the author should first characterize whether the sorted SOX9^{high}/KRT20^{low} cells possess stem-like molecular characteristics. This may be done using RNA-sequencing of SOX9^{high}/KRT20^{low} versus SOX9^{low}/KRT20^{high} cell populations and examine the enrichment of stem-related signatures.
2. The validation of SMARCB1 as a CRC stem regulator is flawed by the fact that much of data are growth related. While cancer cell differentiation, to some degree, is associated with tumor regression and growth retardation, the data demonstrating a direct role in CRC stemness and differentiation is lacking. The characterization of marker gene expression should go beyond SOX9 and RKT20, and include other well-characterized intestinal stem and differentiation markers, such as LGR5, ASCL2, Olfm4 and CEACAM1. IHC analysis of these markers in xenograft sections may be helpful to clarify the differentiation phenotype following SMARCB1 knockdown.

3. Mechanistic interpretation on how SMARCB1 regulate CRC stemness needs to be provided. This is important since a significant number of SWI/SNF chromatin remodeling complex subunits were identified from the epigenetic-focused library screen. Does SWI/SNF complex directly regulate SOX9 expression in CRC? Does this regulation involve any stem-related signaling pathways, such as Wnt/ β -catenin or Hippo/Yap? The reviewer believes that RNA-seq and further bioinformatic analysis of gene signatures may be required to clarify the signaling pathways involved. These signaling events should be also validated to clarify the mechanism underlying the role of SMARCB1 or SWI/SNF complex in stem regulation.

Minor comments

1. Please clarify the use of RNA interference, instead of CRISPR knockout, as the intervening approach for functional validation in Figures 4-5, given the fact that SMARCB1 was identified using CRISPR library screening?

2. Figure 4h. The normalization of mRNA expression is not clear. Are these mRNA normalized to NTC group? In that case, it is necessary to include NTC and shSMARCB1#1 (without SOX9 overexpression) in this data.

3. Figure 5a-b is hard to follow. There is no direct association between proliferation scores and differentiation. Stem cell markers, such as LGR5 and ASCL2, may have little to no growth effects in CRC. Thus, the data from Depmap is weak to support the roles in CRC stemness.

4. The morphological data presented in Figure 5e is not convincing. Because of insufficient resolution and lack of differentiation markers, it is unclear whether the organoids in SMARCB1 knockdown condition have crypt outgrowth. Immunofluorescent staining of differentiation markers should be used to show evidence of organoid differentiation.

5. Figure 5g shows KRT20 is absent in protein level in CRC model. As ~85% of CRCs express KRT20 (CK20 is a pathological marker for CRC), the authors are advised to test additional organoid/PDX models with more physiologically relevant levels of basal KRT20. Also, a histopathological analysis of these tumors post-treatment would be useful to assay morphological evidence of differentiation.

Reviewer #2 (Remarks to the Author):

This work by Spisak and team is noteworthy in demonstrating that a dual reporter system is more powerful for performing dropout screens than single reporter systems that are more commonly used. The study also links some new epigenetic regulators to colon cancer progression.

The work begins with a comparison of single-reporter colon cancer cell lines and identifies genes with epigenetic regulatory functions in a targeted Crispr screen using a fluorescent reporter for Sox9 expression. Next, a similar screen is conducted but with a reporter marking cellular differentiation in the colon, Krt20.

In Fig. 3, the authors then use the dual reporter, which is essentially a double transgenic cell line using the first two reporters. The authors conduct a Perturb-seq study to further investigate transcriptional shifts caused by knockdown/knockout of epigenetic regulators and find that SMARCB1, KMT2A, and SMARCA4 are suppressors of differentiation, much like SOX9 and CTNNB1, which were found and served as positive controls.

The authors follow up showing that SMARCB1 is important for SOX9 expression and suppression of KRT20 expression using organoids and cancer cell lines via SMARCB1 shRNA. They also mine Dependency Map data to further link SMARCB1 to a function as a colon tumor suppressor, and finally show that shRNA targeting of SMARCB1 leads to suppressed growth of colon cancer cell xenografts or colon cancer organoids.

The work is certainly of interest in that it provides new screening technology and identifies candidate epigenetic regulators of colon cancer progression. Some limitations include the lack of explanation for some experimental choices, and uncertain mechanism of action of the hits from the screen.

Some minor points:

There seems to be a mix of cell lines and organoids, shRNAs and gRNAs. There didn't seem to be a rationale for switching between systems and cell lines in many cases. What was the rationale for using the chosen cell lines and why were some assays done in certain cell lines or organoids and not others?

In Figure 3e and 3f, an overlap analysis shows genes found that were in common between different screens, however, some of the genes are found in multiple portions of the venn diagram (occurring more than once within a condition). For example, DOT1L appears as both a consensus gene across all screens but also as a gene only found in the dual reporter screens. I believe this must be an error.

The literature appears mixed on SWI/SNF restricting differentiation and acting as an oncogene, and the functions could certainly be context specific. It would be exciting to see if over-expression would promote growth, or what the consequences of SMARCB1 knockout would have on gene regulation.

In discussion paragraph 1, replace "surfacr" with "surface".

Reviewer #3 (Remarks to the Author):

Spisak et al report on a series of endogenous locus reporters and their use in exploring the differentiation of colorectal cancer.

The work is of potential interest to members of the field, and appears to show some interesting, if modest, phenotypic effects.

The work could be strengthened by some controls that could help distinguish generalizable from specific/artifactual results.

- The manuscript initially describes the generation and characterization of a LS180 SOX9-GFP cell line (Fig 1 b-f). Then rather abruptly switches to an HT29 SOX9-mKate line, without sharing characterization of this line. The reason for this switch is not evident. Especially given that LS180 results in focused knockout screening are not very striking/consistent (supp fig 2), it doesn't seem like this cell line is a good basis for benchmarking the method. Instead HT29 should be used throughout Fig1.
- Especially in the case of SOX9 which can causally affect cell phenotype, efforts should be made to assess the effects of the knockin reporter on gene function: expression level, message and protein half-life, should be assessed to clarify how much the observed results are perturbed by the method of measurement.
- The observed effects on SOX9 and KRT20 reporters should be correlated to measurement of the respective proteins' abundance via western blot or FACS.
- In figures 1 and 2, the comparison is made between the bottom 2.5% and top 2.5% of GFP. Whereas the bottom 2.5% has an interpretation in terms of depletion of the reporter, the top 2.5% does not, the comparison should be made to the average of the negative control treatments. The comparison to the top 2.5% stochastically overexpressed GFP just serves to increase the apparent signal 'window.'
- Figure 4e does not show a SMARCB1 knockout western control.
- Figure 4g and 4h should both have measurements of SMARCB1, KRT20, and SOX9 (i.e. SOX9 levels should be added to 4f and KRT20 to 4g).

Other comments:

- Page 4 and page 6 both have the phrases: 'As expected, GFP and SOX9/KRT20 sgRNAs and shRNAs were consistently depleted in the GFP low fraction.' However from my reading this is the opposite of what is seen: these are depleted in the GFP high fraction.

- The authors do not motivate the particular focused CRISPR library. Even if their interest lies in epigenetic targets, and a full-genome screen is beyond the scope of this work, it could certainly be argued that additional known CRC, WNT, cell cycle, etc targets should have been included in order to benchmark the effect sizes of the epigenetic regulators to known pathway players, and also to bridge to other published perturbation screens referenced.
- The knockdown efficacy is worryingly low in Fig 1c, 5c, and 5e. 50% knockdown is in most cases not biologically relevant as evidenced by the prevalence of heterozygous loss of function mutations at most loci in the 'normal' human population: a 50% dosage difference is in most cases phenotypically invisible. The authors should justify why such modest effects would be expected to have meaningful outcomes.

Reviewer #1 (Remarks to the Author):

In this manuscript, Sandor Spisak and colleagues utilized single and dual fluorescent reporters of endogenous genes to investigate CRC stemness and differentiation phenotypes. They performed CRISPR screening using an epigenetic focused library and identified multiple stem-regulating candidates. Perturb-seq validation indicated that SMARCB1, a subunit of the SWI/SNF BAF complex, is a negative regulator of CRC differentiation. Finally, using Broad's Depmap portal and xenograft models, they revealed that SMARCB1 is required for the growth of CRC tumors. Interference of SMARCB1 caused upregulation of KRT20 at both mRNA and protein levels, suggesting a differentiation phenotype in these xenografts.

While the authors provide a sophisticated dual reporter system in combination with epigenetic-focused CRISPR library screening to dissect CRC stemness regulation, the utilization of dual reporter to monitor CRC stem and differentiated cells itself is not novel (Mariko Shimokawa et al. *Nature*. 2017 May 11;545(7653):187-192). Moreover, the fidelity of reporter system as the indicator of CRC stem cells should be validated using more intensive molecular and phenotype characterization, before conducting CRISPR library screening. The validation of SMARCB1 as a CRC differentiation factor lack mechanistic insights. The molecular events leading to CRC differentiation and SOX9/KRT20 expression alterations need to be clarified. In general, while this study provides some interesting methodological and mechanistic insights into CRC stemness regulation, its novelty is hampered by insufficient molecular characterization and mechanistic depth. The manuscript may be considered for publication pending the satisfaction of following comments:

We thank the reviewer for summarizing and contextualizing our study. We appreciate the study by the Sato lab referenced above and believe it represents an important example of the reciprocal nature between stem cell (in their case *Lgr5*) and differentiation (*Krt20*) in CRC. The team pursues a more technically challenging system in patient-derived organoids and showed the functional importance of *Lgr5*⁺ cells in human CRC using an ablation system. There are a few distinctions worth mentioning: (1) their knock-in reporters were established in CRC organoid clones, whereas our system is engineered into the population of human CRC cell lines; (2) they utilized single-sided reporters and combine with *in situ* evaluation, whereas we eventually utilize a dual endogenous reporter (knock-in of both reporters into the same cell/population); (3) they utilize their system for lineage tracing, dependency of *Lgr5*, and demonstrating CRC plasticity, whereas we utilize the dual reporter system for functional CRISPR screens to nominate new regulators; and (4) while *Lgr5* is an important normal stem cell marker, there is now evidence that stem cell-like populations in CRC have markers of regeneration and fetal reprogramming (PMID:36332574), with our group focusing on SOX9's role in fetal reprogramming CRC (PMID:36989360). Perhaps the reason we didn't mention this study in the introduction is because we were focused on reporter systems to which CRISPR screens were applied. We have now cited this important study in the introduction and discussion.

1. A drawback of this study is the use of CRC cell lines as models of CRC stemness. Monolayer cells lack the microenvironment of stem cell niche and thus may exhibit distinct feature with cancer stem cells *in vivo*. Because of this difference, the author should first characterize whether the sorted SOX9^{high}/KRT20^{low} cells possess stem-like molecular characteristics. This may be done using RNA-sequencing of SOX9^{high}/KRT20^{low} versus SOX9^{low}/KRT20^{high} cell populations and examine the enrichment of stem-related signatures.

We appreciate this point raised by the reviewer. For the revised manuscript, we performed bulk RNA-sequencing on flow sorted cell populations from HT-29 CRC with SOX9^{mK2} and KRT20^{GFP} endogenous reporters. SOX9^{mK2} high/KRT20^{GFP} low demonstrate elevated expression of stem cell genes as indicated by *SOX9*, *ASCL2*, *CEACAM1* and *LGR5* (Reviewer-only Figure 1a). By contrast, KRT20^{GFP} high/SOX9^{mK2} low sorted fraction demonstrated elevated expression of differentiation markers as indicated by increased *KRT20*, *MUC2* and *DPP4* mRNA levels (Reviewer-only Figure 1b). Beyond these individual markers, gene set enrichment analysis (GSEA) of mRNA profiles from SOX9^{mK2} high/KRT20^{GFP} low fraction showed enrichment of stem cell signatures, whereas the opposite fraction showed enrichment of intestinal differentiation signatures (Reviewer-only Figure 2). We now display this result in Figure 3 using a composite GSEA plot of all 4 signatures (Fig. 3b) and select markers in (Fig. 3c). We will release the raw data in the GEO repository under the accession number GSE236257 to further support the reproducibility of our work.

a Stem Cell Markers

b

Differentiation Markers

Reviewer-only Figure 1. Expression of stemness (a) and differentiation (b) markers in the flow-sorted GFP positive compared to mKate2 positive fractions of the HT-29^{SOX9-mKate2/KRT20-GFP} dual reporter cell line.

a Intestinal Stem Cell Gene Signatures

b Intestinal Differentiation Gene Signatures

Reviewer-only Figure 2. Gene set enrichment analysis of intestinal stem cell and differentiated enterocyte gene expression signatures in the flow-sorted GFP positive compared to mKate2 Positive fractions of the HT-29^{SOX9-mKate2/KRT20-GFP} dual reporter cell line.

To further support our system indicating that SOX9^{mK2}-high/KRT20^{GFP}-low express stem-like genes and SOX9^{mK2}-low/KRT20^{GFP}-high express differentiated genes, we re-analyzed public scRNA-seq data of epithelial cells from neoplastic lesions derived from the AOM/DSS CRC mouse model (sourced from GEO at GSM7029393 and GSM7029394) (**Reviewer-only Figure 3**). In this dataset, we observed differential expression of stemness and differentiation markers across epithelial cells, where SOX9^{mK2}-high/KRT20^{GFP}-low cells show high expression of known stem cell genes (*Ascl2* and *Lgr5*) and SOX9^{mK2}-low/KRT20^{GFP}-high cells show high expression of known differentiation genes (*Krt20* and *Muc2*).

Reviewer-only Figure 3. (a) Workflow for analysis of expression of stemness and differentiation markers and gene sets in SOX9-high KRT20-low stem-like epithelial cancer cells compared to SOX9-low KRT20-high differentiated epithelial cancer cells from an AOM/DSS mice model of colorectal cancer initiation. (b) UMAP of epithelial cancer cells from the AOM/DSS mice model colored by normalized expression of stemness and differentiation markers.

Overall, we agree with the reviewer that 2D monolayer culture lacks important features of 3D culture (e.g., organoids), such as stem cell niche. Further, we believe that CRC cell line models are not as faithful as patient-derived CRC organoids. Our idea was to engineer the 2D cell line system, which was more feasible for generating a dual reporter, then perform genetic screens, which also yields higher quality results in cell lines than organoids due many technical reasons, and then validate our results in both cell line and patient-derived organoid models, which we did for SMARCB1.

2. The validation of SMARCB1 as a CRC stem regulator is flawed by the fact that much of data are growth related. While cancer cell differentiation, to some degree, is associated with tumor regression and growth retardation, the data demonstrating a direct role in CRC stemness and differentiation is lacking. The characterization of marker gene expression should go beyond SOX9 and KRT20, and include other well-characterized intestinal stem and differentiation markers, such as LGR5, ASCL2, Olfm4 and CEACAM1. IHC analysis of these markers in xenograft sections may be helpful to clarify the differentiation phenotype following SMARCB1 knockdown.

We appreciate the reviewer's suggestions and apologize for providing an impression that we mainly focused on anti-proliferation effect of SMARCB1. We tried to focus Figure 4 on stem cell/differentiation effects of SMARCB1 loss and Figure 5 on the anti-tumor effects.

Enterocyte-specific genes (Wang et al., 2020)

Reviewer-only Figure 4. RNA-seq profiling of patient-derived CRC organoid upon SMARCB1 knockdown versus nontargeting control (NTC). Gene set enrichment analysis (a), volcano plot (b), and heatmap (c) of differentiated enterocyte gene expression signature. Volcano plot displays all genes in the differentiated enterocyte signature in green. Heatmap shows up-regulated genes in the signature (LFC > 0.5).

The PERTURB-seq experiment in **Fig. 4b-d** demonstrates that SMARCB1 knockout in HT-29 CRC cells leads to induction of differentiation and loss of stem cell signaling based on broad gene expression signatures. To extend these results based on reviewers' suggestions, we performed bulk RNA-sequencing on patient-derived CRC organoids following SMARCB1 KD and found a robust induction of differentiated enterocyte gene signatures (**Fig. 4g** and **Reviewer-only Figure 4a-c**). This was also supported by IHC of KRT20, which showed strong induction, and, to a lesser extent, differentiation marker DPP4 in SMARCB1 KD xenografts (**Fig. 5f** and **Reviewer-only Figure 5**). The induction of DPP4 is better depicted by immunoblot (**updated Fig. 5g**). However, unlike the PERTURB-seq data in HT-29 CRC cells, which showed downregulation of stem cell programs, SMARCB1 KD in patient-derived CRC organoids had a modest negative effect on stem cell markers by IHC, showing a modest downregulation of LGR5 and no effect on SOX9 (**Fig. 5f** and **Reviewer-only Figure 5**).

Reviewer-only Figure 5. Immunohistochemistry analysis of SMARCB1 and KRT20 proteins in a patient-derived CRC organoid *in vivo* xenograft in an immunodeficient mouse (#23) upon non-targeting shRNA (left) or SMARCB1 sh#4 knockdown (right).

Reviewer-only Figure 5 continued. Immunohistochemistry analysis of DPP4 and LGR5 proteins in a patient-derived CRC organoid *in vivo* xenograft in mouse (#23) upon non-targeting shRNA (left) or SMARCB1 sh#4 knockdown (right).

Based on these results, we revisited our dual reporter data, asking whether SMARCB1 sgRNAs were preferentially enriched in the differentiation reporter. We found that SMARCB1 sgRNAs were enriched in the KRT20^{GFP} high fraction and depleted in the SOX9^{mKate2} low fraction, suggesting contributions from both sides (**Reviewer-only Figure 6**). Taken together, we conclude that all models showed that SMARCB1 restricted differentiation but only our cell line models captured its ability to support stem cell behavior, likely due to the resolution afforded by single cell analysis in the PERTURB-seq data (which may be diluted in bulk analyses like RNA-seq of patient-derived organoids). We therefore tone down the language that SMARCB1 promotes stem cell behavior and maintain our claim that it restricts differentiation, which may be decoupled in CRC.

Reviewer-only Figure 6: SMARCB1 Beta score in the GFP vs. mKate2, GFP vs. Dual Negative, and mKate2 vs. Dual Negative flow-sorted fraction comparisons of the HT-29 SOX9-mCherry KRT20-GFP dual reporter cell line.

3. Mechanistic interpretation on how SMARCB1 regulate CRC stemness needs to be provided. This is important since a significant number of SWI/SNF chromatin remodeling complex subunits were identified from the epigenetic-focused library screen. Does SWI/SNF complex directly regulate SOX9 expression in CRC? Does this regulation involve any stem-related signaling pathways, such as Wnt/ β -catenin or Hippo/Yap? The reviewer believes that RNA-seq and further bioinformatic analysis of gene signatures may be required to clarify the signaling pathways involved. These signaling events should be also validated to clarify the mechanism underlying the role of SMARCB1 or SWI/SNF complex in stem regulation.

We share the desire to understand the mechanism by which SMARCB1 restricts differentiation. We reanalyzed our PERTURB-seq data, analyzing single cell transcriptional profiles of SMARCB1 perturbed cells. In line with our previous results, we found WNT pathway downregulation in SMARCB1 KO cells (**Reviewer-only Figure 7**); JAK-STAT signaling was also downregulated. We also found upregulation of Hypoxia, Trail, EGFR, and TGF β signaling, which are candidate pathways driving differentiation (**Reviewer-only Figure 7**). In particular, TGF β signaling has been shown to promote differentiation in normal intestines (PMID: 30988513) and disruption of the pathway via mutations in its receptor or mediators (e.g. *SMAD4*) is a hallmark of CRC. Bulk RNA-sequencing on SMARCB1 KD patient-derived CRC organoids confirmed downregulation of JAK-STAT signaling and upregulation of Hypoxia but did not capture the other features revealed by PERTURB-seq, which once again may be due the difference in resolution of single cell vs bulk analyses. The validation of these candidates and functional studies to glean a deeper mechanistic understanding of how SMARCB1 restricts differentiation in CRC will require further investigation. We add a clause to study limitations indicating this (page 15). We will certainly be working on this moving forward and hope other labs are stimulated to do the same. For this study, we aimed to introduce the dual reporter system, demonstrate its robustness, nominate functional regulators, and validate at least one.

Reviewer-only Figure 7. Pathway (a) and Transcription Factor activity (b) analyses in PERTURB-seq data in HT29 cell line. Positive score indicates stronger activity in SMARCB1-KO cells versus non-targeting control (NTC) and vice versa.

PMID: 36699385

Badia-i-Mompel P., Vélez Santiago J., Braunger J., Geiss C., Dimitrov D., Müller-Dott S., Taus P., Dugourd A., Holland C.H., Ramirez Flores R.O. and Saez-Rodriguez J. 2022. decoupleR: Ensemble of computational methods to infer biological activities from omics data. *Bioinformatics Advances*. <https://doi.org/10.1093/bioadv/vbac016>

There are some data supporting a role of SWI/SNF chromatin remodeling complex in regulating SOX9. ARID1A, one of the BAF subunits, when lost, resulted in loss of intestinal stem cells in Villin-Cre;Arid1a^{fl/fl} mouse model through regulation of SOX9 (PMID: 30635419). A separate project in our laboratory is looking into the physical interaction between SOX9 and BAF complex members. This line of investigation may reveal additional mechanisms that may not be appreciated by transcriptional profiling.

Minor comments

1. Please clarify the use of RNA interference, instead of CRISPR knockout, as the intervening approach for functional validation in Figures 4-5, given the fact that SMARCB1 was identified using CRISPR library screening?

We understand the reviewer's concern. Since the CRISPR knockout depends on repair following a double strand break, there is a preference for selection of in-frame mutations that preserve function of important genes, like SOX9 and SMARCB1 in CRC, to ensure cell survival. While targeting sgRNAs can be captured by FACS-based screens due to dropout (i.e., loss of sgRNAs that successfully deleted SMARCB1), these cells will be negatively selected against and reduced in cell culture experiments, leading to less effective manipulation in engineered cell lines. We therefore favored shRNA KD experiments which suppresses expression to 75-90% without the strong selective pressure exerted by attempts at knockout. We reduce the chances of off-target effects by using multiple hairpins. Please also see Reviewer-only Figure 10 and discussions below.

2. Figure 4h. The normalization of mRNA expression is not clear. Are these mRNA normalized to NTC group? In that case, it is necessary to include NTC and shSMARCB1#1 (without SOX9 overexpression) in this data.

We apologize for the confusion. All RT-PCR data is normalized to *GAPDH*. The requested data is in Supplementary Fig. 4g.

3. Figure 5a-b is hard to follow. There is no direct association between proliferation scores and differentiation. Stem cell markers, such as LGR5 and ASCL2, may have little to no growth effects in CRC. Thus, the data from Depmap is weak to support the roles in CRC stemness.

Our hypothesis is that inducing CRC differentiation decreases proliferation and reduces tumor growth. Previous work has shown that reintroduction of wildtype Apc in CRC models can promote differentiation and reduce tumor burden (PMID: 26091037, 7882361). Furthermore, depletion of Lgr5⁺ cancer stem cells has a negative impact on CRC tumor growth and metastasis (PMID: 28358093). Our group showed that disrupting SOX9 promotes differentiation and reduces tumor burden across several model systems (PMID: 34571027, 36989360), inspiring the creation of the dual reporter system. To investigate that translational potential of SMARCB1 and its ability to restrict differentiation, we focus Figure 5 on anti-tumor effects, adding additional data in the revision to demonstrate induction of differentiation upon SMARCB1 loss. Indeed, many of the top genes that once perturbed induced differentiation are also dependencies in many CRC cell lines (Figure 5a). Taken together, identifying factors that prevent differentiation may be strong therapeutic candidates as their perturbation can reduced tumor growth and cell state plasticity (the latter is not addressed in this study).

To avoid confusion in main figures, we moved Fig. 5b to the supplement and combined it with Supplementary Fig. 5a.

4. The morphological data presented in Figure 5e is not convincing. Because of insufficient resolution and lack of differentiation markers, it is unclear whether the organoids in SMARCB1 knockdown condition have crypt outgrowth. Immunofluorescent staining of differentiation markers should be used to show evidence of organoid differentiation.

We agree that the morphological assessment in Figure 5d does not specifically assess differentiation; we therefore removed the clause "consistent with differentiation". In the revision, we pursued IHC of the xenografts from the same patient-derived CRC organoid to provide stronger evidence of differentiation induction (Fig. 5f). Furthermore, GSEA analysis of bulk RNA-seq data in Fig. 4g now demonstrates the activation of a broad differentiation program upon SMARCB1 KD in these organoids.

5. Figure 5g shows KRT20 is absent in protein level in CRC model. As ~85% of CRCs express KRT20 (CK20 is a pathological marker for CRC), the authors are advised to test additional organoid/PDX models with more

physiologically relevant levels of basal KRT20. Also, a histopathological analysis of these tumors post-treatment would be useful to assay morphological evidence of differentiation.

We performed a SMARCB1 loss-of-function assay on a variety of models, including 2 cancer cell lines (HT29 and HT115), human adenoma and patient-derived CRC organoids. All of them were tested in-vitro with some tested in a xenograft model. All models consistently showed induction of KRT20 upon SMARCB1 loss (see table below). The patient-derived CRC organoid xenograft also expressed KRT20 in 1-5% of cell in each crypt (Reviewer-only Figure 5)

Models	In-vitro	In-vivo	KRT20	SOX9
CRC Cell line – HT29	Yes	Yes	Strong Increase	Decrease
CRC Cell line – HT115	Yes	No	Increase	Decrease
Patient-derived adenoma organoid	Yes	No	Increase	Modest Decrease
Patient-derived CRC organoid	Yes	Yes	Strong Increase	No change

Reviewer #2 (Remarks to the Author):

This work by Spisak and team is noteworthy in demonstrating that a dual reporter system is more powerful for performing dropout screens than single reporter systems that are more commonly used. The study also links some new epigenetic regulators to colon cancer progression.

The work begins with a comparison of single-reporter colon cancer cell lines and identifies genes with epigenetic regulatory functions in a targeted Crispr screen using a fluorescent reporter for Sox9 expression. Next, a similar screen is conducted but with a reporter marking cellular differentiation in the colon, Krt20. In Fig. 3, the authors then use the dual reporter, which is essentially a double transgenic cell line using the first two reporters. The authors conduct a Perturb-seq study to further investigate transcriptional shifts caused by knockdown/knockout of epigenetic regulators and find that SMARCB1, KMT2A, and SMARCA4 are suppressors of differentiation, much like SOX9 and CTNNB1, which were found and served as positive controls.

The authors follow up showing that SMARCB1 is important for SOX9 expression and suppression of KRT20 expression using organoids and cancer cell lines via SMARCB1 shRNA. They also mine Dependency Map data to further link SMARCB1 to a function as a colon tumor suppressor, and finally show that shRNA targeting of SMARCB1 leads to suppressed growth of colon cancer cell xenografts or colon cancer organoids.

The work is certainly of interest in that it provides new screening technology and identifies candidate epigenetic regulators of colon cancer progression. Some limitations include the lack of explanation for some experimental choices, and uncertain mechanism of action of the hits from the screen.

We appreciate the summary of our results and overall positive assessment. In the revised manuscript, we attempt to better explain our experimental choices and nominate possible mechanisms.

Some minor points:

There seems to be a mix of cell lines and organoids, shRNAs and gRNAs. There didn't seem to be a rationale for switching between systems and cell lines in many cases. What was the rationale for using the chosen cell lines and why were some assays done in certain cell lines or organoids and not others?

We appreciate the concern and attempt to be more transparent for our choices in the revised manuscript. Here is a summary in the order they appear in the manuscript:

1. The reporter system was engineered in 3 CRC cell lines (HT-29, HT-115, LS180). While SOX9 reporter performance was strong in all 3, KRT20 reporter worked best in HT-29 and HT-115; LS180 KRT20 reporter did not perform as well, which can be seen in **Supplementary Fig. 3e**: KRT20 sgRNAs and corresponding fluorescent probe sgRNAs have a modest effect size. As a result, we took advantage of the SOX9 reporter performance in LS180 for Figure 1 (first engineered line we created and most comprehensive validation data with genomic integration validation and SOX9 shRNA experiments). We also show a SOX9 only reporter in HT-29 using the other fluorescent marker (mKate2), which we now clarify on page 5.
2. In the screening format, we tested both shRNAs and sgRNAs. There was a clear advantage to sgRNAs as the data in Figure 1 and 2 indicated in the screens. However, when it comes to validation experiments in Figures 4 and 5, shRNAs were the method of choice due to reasons alluded to in a response to Reviewer #1. Briefly, CRISPR KO does not perform well for essential/important genes due to selection of in-frame mutations that presumably preserve protein function; we faced this while studying SOX9 (PMID: 34571027, Figure 2H). We noticed the same while studying SMARCB1 (please see **Reviewer-only Figure 10**). We therefore favored shRNA-mediated suppression since phenotypes and molecular data can be captured following 75-90% knockdown.
3. Finally, in Figures 4 and 5, we tried to utilize multiple CRC models beyond cell lines to validate our results and demonstrate their impact. In addition to HT-29 and HT-115 CRC cell lines, we used two patient-derived organoids – one from a patient with familial adenomatous polyposis, so a premalignant adenoma, and another from a patient with CRC carrying APC and KRAS mutations.

In Figure 3e and 3f, an overlap analysis shows genes found that were in common between different screens, however, some of the genes are found in multiple portions of the venn diagram (occurring more than once within a condition). For example, DOT1L appears as both a consensus gene across all screens but also as a gene only found in the dual reporter screens. I believe this must be an error.

To clarify, the overlap analysis shown in previous Figures 3e and 3f (now **Figure 3g and 3h**) shows the count of enriched and depleted individual gRNAs rather than genes. Since there are 6 gRNAs per gene, some of the gRNAs targeting the same gene can be found in different compartments of the Venn Diagram. We understand that this could be confusing, but it is intended to show guide distribution and gradients of gene inactivation.

For example, the observation that DOT1L appears as a duplicated gene in the consensus set and the Dual Reporter: High GFP vs Low mKate2 set in **Figure 3g** is due to

- 1) 2 DOT1L gRNA were enriched in GFP across all 3 comparisons so there is a DOT1L gene annotation in the consensus set of the Venn diagram
- 2) 2 DOT1L gRNA were enriched in GFP in only the 2 Dual Reporter comparisons and not the Single Reporter comparison so there is a DOT1L gene annotation in this subset of the Venn diagram
- 3) 1 DOT1L gRNA was enriched in the GFP Positive vs. mKate2 Positive of the Dual Reporter system only. We excluded this gRNA annotation for visual acuity of the figure, given that DOT1L has already been annotated as a hit with multiple gRNA representation from other comparisons.
- 4) 1 DOT1L gRNA were not enriched in GFP in any comparison and excluded from visualization in this overlap analysis

To further explain this, we prepared **Reviewer-only Figure 8** to show in which systems DOT1L gRNA scored. In summary, the overlap analysis of Figure 3g and 3h represents top-scoring gRNA hits, but multiple gRNA per gene were included in the epigenetic library. Thus, different gRNA targeting the same gene may be enriched or depleted in unique subsets of comparisons due to biological variation in knockout efficiency as expected. Highly confident gene hits of the genetic screen should have consistent enrichment or depletion of multiple gRNA targeting the gene across multiple fraction comparisons. To clarify this observation for future readers, we have uploaded the gRNA-level screen hits to accompany the screen data analysis code at this link.

Single Reporter System

Enriched in GFP High vs. GFP Low

Dual Reporter System

Enriched in GFP Positive vs. Dual Negative

Reviewer-only Figure 8. Overlap analysis of which of the 6 DOT1L gRNA scored as an enriched hit in each of the 3 different reporter system comparisons.

The literature appears mixed on SWI/SNF restricting differentiation and acting as an oncogene, and the functions could certainly be context specific. It would be exciting to see if over-expression would promote growth, or what the consequences of SMARCB1 knockout would have on gene regulation.

We agree that the function of SWI/SNF is context specific. In neurons, it is recognizable that SMARCB1 suppression leads to impaired neuron differentiation (PMID 31033435, PMID 32912900), whereas its impact on intestinal development is much less studied. There is one study showed that SMARCB1 loss in human pluripotent stem cell resulted in upregulation of differentiation markers partly through WNT pathway (PMID 33125876). As summarized above, we performed single cell and bulk RNA-seq analysis with SMARCB1 perturbation in two CRC models and nominated specific pathways. In line with the effect on stem cell and differentiation programs, we observed downregulation of WNT and upregulation of TGF β signaling (Reviewer-only Figure 7, red boxes). Other pathways implicated include upregulation of TRAIL and HYPOXIA signaling whereas downregulation of JAK-STAT signaling. Whether these pathways functionally participate in differentiation induction requires further investigation. We add a clause to the study limitations section of the discussion that points out the molecular mechanism by which SMARCB1 restricts differentiation requires further elucidation (page 15).

In discussion paragraph 1, replace "surfacers" with "surface".

We apologize for this oversight and corrected the error.

Reviewer #3 (Remarks to the Author):

Spisak et al report on a series of endogenous locus reporters and their use in exploring the differentiation of colorectal cancer.

The work is of potential interest to members of the field, and appears to show some interesting, if modest, phenotypic effects.

The work could be strengthened by some controls that could help distinguish generalizable from specific/artifactual results.

- The manuscript initially describes the generation and characterization of a LS180 SOX9-GFP cell line (Fig 1 b-f). Then rather abruptly switches to an HT29 SOX9-mKate line, without sharing characterization of this line. The reason for this switch is not evident. Especially given that LS180 results in focused knockout screening are not very striking/consistent (supp fig 2), it doesn't seem like this cell line is a good basis for benchmarking the method. Instead HT29 should be used throughout Fig1.

In general, we agree with the reviewer's assessment. As summarized above, we provide a better explanation for our choice of models and add language on page 5 to better explain reasons to switch lines. Since LS180 was the first engineered SOX9 reporter line created, it underwent the most comprehensive evaluation. In fact, its performance as a SOX9 reporter is similar to HT-29 and HT-115 (**Fig. 1f-g and Supplementary Fig. 3e-g**). It suffered when functioning as a KRT20 reporter, which is why we use the HT-29 engineered line for most of the manuscript. HT-29 was the best model for dual reporter activity and, although included in the main figure and supplement for the single-sided SOX9 reporter, it is featured in the main figure of all remaining results, including single-sided KRT20 reporter (**Fig. 2**), dual reporter (**Fig. 3**), and validation studies in **Figs. 4 and 5**. . Furthermore, we performed molecular validation of the HT-29 reporter system in **Figure 3b-c** by RNA-seq evaluation of flow-sorted populations, hopefully providing the benchmarks to trust this system.

Supplementary Fig. 2 is focused on KRT20 reporter and does not feature LS180.

- Especially in the case of SOX9 which can causally affect cell phenotype, efforts should be made to assess the effects of the knockin reporter on gene function: expression level, message and protein half-life, should be assessed to clarify how much the observed results are perturbed by the method of measurement.

We appreciate the reviewer's concerns regarding technical confounding due to the knock-in. The T2A sequence separates the fluorescent knock-in from the protein after translation, hence the fluorescent reporter should in practice not impact SOX9 function (**Figs. 1a and 2b**).

Biased examination of the phenotypes of the engineered vs. parental HT29 cell lines showed that the cell lines have no difference in proliferation behavior. SOX9 knockdown/knockout induces differentiation and KRT20 in the engineered cell line, as observed in the parental line.

In addition, we measured the transcriptomic profile of the HT29 dual endogenous reporter system cell line using RNA-seq. Here, we observed that both the global gene expression and the differentiation/stemness marker-specific expression of HT29 from the Cancer Cell Line Encyclopedia (CCLE) and HT29-engineered reporter system cell line are positively correlated ($R=0.849$, $P<0.001$) (**Reviewer-only Figure 9**). Taken together, we observed no transcriptomic nor gross phenotypic changes due to the knock-in at the endogenous genomic locus.

Reviewer-only Figure 9. Spearman ranked correlation of ranked gene expression between parental HT29 cell line (CCLE) and HT29 SOX9-mK2 and KRT20-GFP dual reporter cell line.

- The observed effects on SOX9 and KRT20 reporters should be correlated to measurement of the respective proteins' abundance via western blot or FACS.

The best experiment indicating the correlation of SOX9 expression with GFP expression is by FACS in **Figure 1c**. SOX9 KD corresponded to reduction in GFP+ LS180 cells. The hairpins were validated in our previous study (PMID: 34571027). Copied below for convenience.

In addition, the screens indicate that shRNA or sgRNA mediated disruption of SOX9 and corresponding fluorescent probe as well as KRT20 and corresponding fluorescent probe are found in the correct FACS sorted compartments (i.e., SOX9-GFP reporter, both SOX9 and GFP sgRNAs are found in the GFP low/negative fractions). We believe these perturbation correlations are stronger than those by endogenous expression alone.

- In figures 1 and 2, the comparison is made between the bottom 2.5% and top 2.5% of GFP. Whereas the bottom 2.5% has an interpretation in terms of depletion of the reporter, the top 2.5% does not, the comparison should be made to the average of the negative control treatments. The comparison to the top 2.5% stochastically overexpressed GFP just serves to increase the apparent signal 'window.'

We agree that the comparison of the bottom 2.5% and top 2.5% shows the most extreme comparisons of the reporter system to increase the power to discriminate the top enriched and depleted hits of the genetic screen.

In **Figs. 1g and 2e**, we show stepwise comparisons between 25-50%, 50-75%, and 75-100% fractions to the 2.5%. Here, we show a gradual increase in discriminatory power based on enrichment of gRNA targeting experimental control genes including SOX9, KRT20, GFP, and mKate2 as we compare more different sorted fractions based on reporter fluorescence. Thus, the comparison of the bottom 2.5% and top 2.5% only serves to increase the discriminatory power of the genetic screen using the endogenous reporter system given that the same trends in enrichment and depletion of experimental control genes always holds true regardless of the fraction comparison.

• Figure 4e does not show a SMARCB1 knockout western control.

We apologize for not providing this in the original manuscript. As alluded to above, CRISPR/Cas9 knockout has been difficult to achieve for important genes, like *SMARCB1*. We provide an immunoblot in **Reviewer-only Figure 10**, which includes the **Figure 4e** immunoblot for single clones (a), the sanger sequencing of PCR product at cut-site of many clones tested (b), and immunoblot of stably selected shRNA (left) and sgRNA (right) HT-29 cell line populations (c). While original clones chosen did not show a clear loss of SMARCB1 protein (E10 shows a modest decrease), they demonstrated the greatest editing frequency (**Reviewer-only Figure 10a-b**). We grew additional slow-growing clones from the *SMARCB1* Cas9/sgRNA populations during the revision period but continued to find likely heterozygous alterations (**Reviewer-only Figure 10b**). In the revision, we amend the figure legend to include the sanger sequence editing frequency for each clone and the text to indicate partial knockout. We are open to other suggestions to present this data and will consider removing it if distracting to the overall results/conclusions.

Reviewer-only Figure 10. (a) Immunoblot analysis of 2 single clones derived from a population of HT29 that underwent CRISPR-editing against *SMARCB1* gene locus versus a parental clone. (b) Table displays the percentages of editing efficiency and in-frame mutation (multiples of 3 bases) as determined by TIDE analysis of Sanger sequencing traces at the targeted genomic locus of *SMARCB1*.

• Figure 4g and 4h should both have measurements of SMARCB1, KRT20, and SOX9 (i.e. SOX9 levels should be added to 4f and KRT20 to 4g).

We added SOX9 levels to 4f, which indicate *SMARCB1* KD leads to the consistent 50% reduction in SOX9 transcript levels. That is validated in 4g. The goal of 4h is to focus on how *SMARCB1* loss affects SOX9 transcripts. The KRT20 induction upon *SMARCB1* KD/KO in the same CRC cell line is provided in 4e.

Other comments:

• Page 4 and page 6 both have the phrases: 'As expected, GFP and SOX9/KRT20 sgRNAs and shRNAs were

consistently depleted in the GFP low fraction.' However from my reading this is the opposite of what is seen: these are depleted in the GFP high fraction.

We apologize for the lack of clarity in our text description of **Fig. 1e,f** and **Fig. 2c-e** on Page 4 and 6. To clarify, the intended meaning was that GFP and its associated reporter gene were depleted in the GFP high fraction compared to the GFP low fraction, in concordance with the reviewer's interpretation. To address this, we have clarified the text associated with Figure 1e,f and Figure 2c-e on Page 4 and 6 to explicitly note which comparisons are being made between two different fractions

- The authors do not motivate the particular focused CRISPR library. Even if their interest lies in epigenetic targets, and a full-genome screen is beyond the scope of this work, it could certainly be argued that additional known CRC, WNT, cell cycle, etc targets should have been included in order to benchmark the effect sizes of the epigenetic regulators to known pathway players, and also to bridge to other published perturbation screens referenced.

We regret not including more comprehensive control genes as the reviewer suggests. This focused CRISPR screen was designed to be a pilot study for a whole-genome CRISPR knockout screen with the same reporter system, which is a current goal of the laboratory. Based on the results of this screen, we are encouraged to pursue a whole genome screen. As for controls, we believe SOX9 was our strongest biological control as SOX9 sgRNAs were consistently found in the KRT20^{high} FACS populations (**Fig. 2c-g**). We designed the gRNA library based on existing drugs targeting epigenetic regulators; we broadened the library to include family members of these drug targets. We hope to perform drug screens related to this work in the future and relate those results to these genetic screen results.

- The knockdown efficacy is worryingly low in Fig 1c, 5c, and 5e. 50% knockdown is in most cases not biologically relevant as evidenced by the prevalence of heterozygous loss of function mutations at most loci in the 'normal' human population: a 50% dosage difference is in most cases phenotypically invisible. The authors should justify why such modest effects would be expected to have meaningful outcomes.

In short, we believe these results are based on a desire for CRC to retain these genes. The hairpins used in Figure 1c are from our recent publication and demonstrated consistent downregulation of SOX9 as we show above (PMID: 34571027). These hairpins were also sufficient to induce differentiation in Apc^{KO}-Kras^{G12D} organoids grown as xenografts *in vivo* (Figure 4, PMID: 34571027, pasted below for convenience). But, as described in the manuscript, there were two xenografts that escaped SOX9 KD, similar to the two xenografts that escaped SMARCB1 KD (**Supplementary Fig. 5c**). We believe these seemingly modest changes in protein is due to selective pressure to escape downregulation. The phenotypes, however, are quite strong. For example, **new Fig. 5f** shows a remarkable induction of KRT20 upon SMARCB1 KD by IHC of *in vivo* xenografts. We also hope the additional evidence of SMARCB1 sgRNA knockout escape via in-frame mutations in **Reviewer-only Figure 10b** helps explain these results.

Figure 4. SOX9 knockdown promotes intestinal differentiation in organoid models of CRC. (A) mRNA expression heat map of intestinal differentiation (blue) and stem cell (red) markers in control and indicated SOX9 shRNA human colon organoids. (B) Immunoblot (Sox9 and vinculin) of mouse control or indicated Sox9 shRNA *Apc^{KO} Kras^{G12D}* colon organoids. Proliferation by CellTiterGlo; mean \pm SD; Student's t test: *** $P < 0.005$, **** $P < 0.001$. (C) Ki67 IHC quantification of indicated fixed mouse *Apc^{KO} Kras^{G12D}* colon organoids; mean \pm SD; Student's t test: *** $P < 0.005$, **** $P < 0.001$. (D) Sox9, Lgr5, Lrig1, Prom1, Axin2, Ascl2, Krt20 mRNA expression in *Apc^{KO} Kras^{G12D}* colon organoids according to qRT-PCR; mean \pm SD; Student's t test: ** $P < 0.01$, *** $P < 0.005$, **** $P < 0.001$. (E) *Apc^{KO} Kras^{G12D}* Sox9 xenograft schematic. Primary tumor xenograft growth curve and day 30 quantification. Representative images of xenograft tumors. (F) Immunoblot (Sox9 and vinculin) of xenograft tumors. (G) Ki67 and Muc2 immunohistochemistry and Alcian blue–periodic acid Schiff (AB-PAS) staining of xenograft tumors; scale bars $\frac{1}{4}$ 100 mm and 20 mm.

REVIEWERS' COMMENTS

Reviewer #1 (Remarks to the Author):

The authors have done an excellent job in addressing my concerns with new data that bolster the conclusions drawn.

Reviewer #2 (Remarks to the Author):

The authors addressed the majority of my concerns. I'd suggest a supplementary note in the paper explaining the rationale for switching between knockdown methods and cell lines, to help the reader understand the rationale that was explained in the rebuttal letter.

Reviewer #3 (Remarks to the Author):

In their revised manuscript, Spisak et al provide responses for the points made in the original review. Unfortunately, the sum total of the responses is more of an explanation for why the results were underwhelming rather than making the results less underwhelming.

The authors bring up several times in response to reviewers' comments that selection for maintenance of some of the interrogated genes leads to low efficacy of knockout/knockdown. Such issues could be addressed with acute treatments, e.g. tet-inducible shRNA constructs.

More fundamentally, the authors acknowledge that their ultimate goal is a genome-wide screen rather than the presented epigenetic target screen. Furthermore, the authors state that the genome-wide screen is underway. It seems a bit like 'salami-slicing' to attempt to extract two papers from one screen. Given that the results for the proof of concept are equivocal and small in magnitude/significance, perhaps this story would be better presented as one or two figures in the manuscript describing the genome-wide screen.

Reviewer #1 (Remarks to the Author):

The authors have done an excellent job in addressing my concerns with new data that bolster the conclusions drawn.

We appreciate the positive feedback.

Reviewer #2 (Remarks to the Author):

The authors addressed the majority of my concerns. I'd suggest a supplementary note in the paper explaining the rationale for switching between knockdown methods and cell lines, to help the reader understand the rationale that was explained in the rebuttal letter.

We appreciate the positive feedback. We will include a supplementary note to this effect.

Reviewer #3 (Remarks to the Author):

In their revised manuscript, Spisak et al provide responses for the points made in the original review. Unfortunately, the sum total of the responses is more of an explanation for why the results were underwhelming rather than making the results less underwhelming.

The authors bring up several times in response to reviewers' comments that selection for maintenance of some of the interrogated genes leads to low efficacy of knockout/knockdown. Such issues could be addressed with acute treatments, e.g. tet-inducible shRNA constructs.

More fundamentally, the authors acknowledge that their ultimate goal is a genome-wide screen rather than the presented epigenetic target screen. Furthermore, the authors state that the genome-wide screen is underway. It seems a bit like 'salami-slicing' to attempt to extract two papers from one screen. Given that the results for the proof of concept are equivocal and small in magnitude/significance, perhaps this story would be better presented as one or two figures in the manuscript describing the genome-wide screen.

We understand some of the reviewer concerns. We attempted to address all points raised by the reviewer. Our knockdown systems were both stable and inducible, phenocopying one another. We also have biological readouts that the knockdown led to molecular and phenotypic changes (e.g., inducing differentiation). For these reasons, we feel confident in the system used for genetic manipulation.